# Hierarchy Decoding: A Training-free Parallel Decoding strategy for Diffusion Large Language Models

**Xiaojing Qi[1,2*]   Lun Du[2*†]   Xingyuan Zhang[3,2]   Lanning Wei[2]   Tao Jing[1]   Da Zheng[2†]**
[1]School of software, Tsinghua University    [2]Ant Group
[3]School of Data Science & Engineering, East China Normal University
qxj24@mails.tsinghua.edu.cn    dulun.dl@antgroup.com
51265903009@stu.ecnu.edu.cn    weilanning.wln@antgroup.com
jintao16@mail.tsinghua.edu.cn    zhengda.zheng@antgroup.com

## Abstract

The utilization of large language models (LLMs) has become increasingly widespread, and has attracted considerable attention. Although Discrete Diffusion Large Language Models (dLLMs) mitigate the inference latency inherent in autoregressive LLM decoding, their computational overhead remains substantial. To address this challenge, we propose **Hierarchy-dLLM**, a hierarchical decoding framework inspired by the divide-and-conquer principle. Our method recursively partitions masked spans into smaller sub-decoding areas and decodes tokens according to their confidence, which substantially increases the number of tokens generated per forward pass and improves information utilization. Extensive experiments conducted on multiple benchmarks demonstrate that Hierarchy-dLLM achieves accuracy comparable to or even surpassing existing baselines. Meanwhile, it is up to $17\times$ faster than vanilla decoding and about $1.5\times$ faster than the Fast-dLLM. These results establish hierarchical decoding as a practical solution for efficient dLLMs inference. The implementation is available at https://github.com/inclusionAI/dInfer.

## 1 Introduction

Although autoregressive (AR) large language models (Radford & Narasimhan, 2018) currently dominate the field, Diffusion large language models (dLLMs) (Yu et al., 2025a) are gaining momentum within the research community due to their unique potential for parallel decoding. In AR decoding, tokens are generated sequentially, which constrains efficiency and limits opportunities for parallelization. In contrast, dLLMs reconstruct linguistic sequences through iterative denoising with bidirectional attention, enabling simultaneous refinement of multiple tokens and thus parallel decoding (Li et al., 2022). Such a property not only improves scalability but also opens new research directions for developing more efficient and flexible decoding strategies.

In practice, however, comparable performance has yet to be observed in the open-source community, despite several commercial closed-source dLLMs claiming impressive throughput (Google DeepMind, 2025; Khanna et al., 2025; Song et al., 2025b). A key reason lies in the architectural trade-off of dLLMs: by adopting bidirectional attention, their per-step computation is significantly slower than that of autoregressive models of similar size. To compensate, dLLMs must achieve substantial gains from parallel decoding. However, representative open-source models such as LLaDA (Zhu et al., 2025) and Dream (HKUNLP, 2025) default to greedy decoding, generating only one token per step. This approach makes their efficiency fall short of AR models, underscoring why accelerating parallel decoding has become a central research focus for dLLMs.

Yet, attempts to scale up parallel decoding face intrinsic difficulties, often referred to as *the curse of parallel decoding* (Wu et al., 2025). This curse arises because tokens predicted within the same

---

[*]Equal contribution.
[†]Corresponding author.

decoding step should satisfy a conditional independence assumption; otherwise, forcing them to be generated simultaneously can lead to substantial performance degradation. For example, given the sentence "*In the classroom, Alice arranged pens, papers, and books neatly on her desk before the teacher began the lesson*", parallel prediction may produce incoherent outputs such as "pens, pens, and pens", illustrating how naive parallel decoding can undermine semantic consistency.

While existing studies primarily focus on confidence-based criteria to determine which tokens should be decoded at each step, such approaches commonly ignore how the spatial distribution of undecoded positions affects the decoding process. To better understand this factor, we conducted a preliminary study and observed that when undecoded tokens are sparsely scattered across the sequence, one-pass decoding produces a token distribution that closely matches step-by-step generation. In contrast, when undecoded tokens form consecutive spans, the resulting distribution exhibits substantial divergence from step-wise generation, leading to pronounced distributional drift. These observations highlight the necessity of incorporating spatial structure into decoding strategies for diffusion-based language models.

To address this challenge, we introduce Hierarchy-dLLM, a novel hierarchical parallel decoding framework inspired by the divide-and-conquer paradigm. Rather than treating all masked positions equally, Hierarchy-dLLM dynamically partitions masked tokens into independent subordinate decoding areas according to the positions of decoded tokens. The proposed decoding strategy is executed independently across individual decoding areas, which allows multiple areas to be decoded in parallel and leads to a significant improvement in overall decoding efficiency.

Our main contributions can be summarized as follows:

1. We performed a comprehensive analysis of the dLLM decoding mechanism. We found that preserving a sparse layout of undecoded tokens within the sequence can effectively reduce distributional drift, thus improving the stability and accuracy of parallel decoding.

2. We propose Hierarchy-dLLM, to the best of our knowledge the first position-based decoding framework for diffusion-based large language models, which systematically leverages divide-and-conquer principles to enhance parallel decoding.

3. We conduct extensive experiments demonstrating that Hierarchy-dLLM achieves superior trade-offs between inference speed and generation quality compared with existing open-source dLLM baselines, running up to $17\times$ faster than vanilla decoding and $1.5\times$ faster than Fast-dLLM.

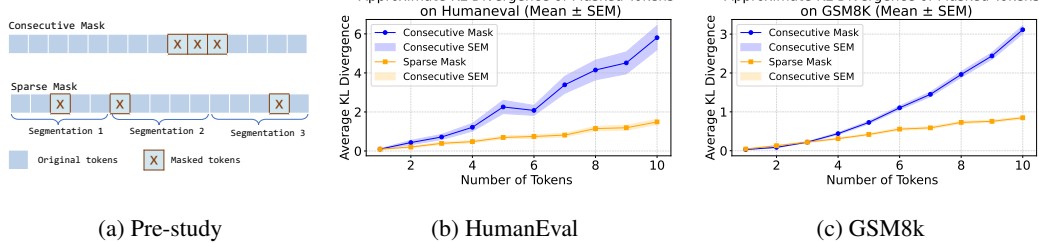

|  |  |  |
|---|---|---|
| (a) Pre-study | (b) HumanEval | (c) GSM8k |

Figure 1: **Average KL-Divergence of Masked Tokens Over Number of Tokens**. (a) shows two masking methods used in our study: Consecutive Mask, where tokens are masked as a contiguous block, and Sparse Mask, where masked tokens are scattered across multiple positions. (b) and (c) report the average KL divergence (mean ± SEM) of masked tokens under these two strategies on HumanEval and GSM8k official answers, respectively. The results indicate that Consecutive Mask generally yields a larger KL divergence than Sparse Mask, suggesting that scattered masking provides more stable token-level predictions across decoding steps.

## 2 PRELIMINARY STUDY

### 2.1 DLLM INFERENCE PROCESS

Within the framework of dLLMs, the current mainstream instantiation is the **Masked Diffusion Model (MDM)** (Shi et al., 2025). We therefore focus our discussion on MDMs in this subsection. Unlike traditional autoregressive models (ARMs) that rely on the chain rule for left-to-right prediction, MDMs construct probabilistic models via masked token prediction, thereby naturally supporting *bidirectional context modeling* and alleviating several limitations of ARMs such as reversal reasoning difficulties and temporal distribution shifts.

**Problem setting.** Let $\mathcal{V}$ denote a fixed vocabulary. We define a sequence $X = (x^1, x^2, \ldots, x^L)$, where each element $x^i \in \mathcal{V}$ represents a token drawn from the vocabulary, and $L \in \mathbb{N}$ denotes the length of the sequence. MDMs define a *forward masking process* that progressively replaces tokens with a special mask symbol $[MASK]$. At time $t \in [0,1]$, the noisy sequence $X_t$ is sampled as

$$q_{t|0}(X_t|X_0) = \prod_{i=1}^{L} q_{t|0}(x_t^i|x_0^i), \quad q_{t|0}(x_t^i|x_0^i) = \begin{cases} 1-t, & x_t^i = x_0^i, \\ t, & x_t^i = M. \end{cases} \tag{1}$$

As $t \to 1$, the sequence becomes fully masked.

**Reverse process.** The reverse process recovers the original data distribution by iteratively predicting masked tokens:

$$q_{s|t}(X_s|X_t) = \prod_{i=1}^{L} q_{s|t}(x_s^i|X_t), \tag{2}$$

where

$$q_{s|t}(x_s^i|X_t) = \begin{cases} 1, & x_t^i \neq M, \; x_s^i = x_t^i, \\ \frac{s}{t}, & x_t^i = M, \; x_s^i = M, \\ \frac{t-s}{t} \, q_{0|t}(x_s^i|X_t), & x_t^i = M, \; x_s^i \neq M, \\ 0, & \text{otherwise.} \end{cases} \tag{3}$$

**Decoding process.** During generation, MDMs start from a fully masked sequence ($t = 1$) and gradually denoise toward $t = 0$. Let $p_0$ denote the prompt, let $r_t$ denote a fully masked sequence, and let $c_i$ denote the confidence score of the $i$-th masked token in $r_t$. Then, the start state of decoding process can be denoted as $X_t = concat(p_0, r_t)$. At each step, the model assigns a predictive distribution over the true values of selected masked tokens:

$$x_s^i = \arg\max p_\theta(X_s \mid X_t), \tag{4}$$

and a proportion $\frac{s}{t}$ of the tokens remain masked according to their confidence, such that only one token is decoded in each step when $\frac{s}{t}$ is scheduled accordingly, and the reverse process remains consistent with the forward process. Importantly, $\frac{s}{t}$ is a tunable parameter that controls the trade-off between speed and fidelity: smaller values correspond to more tokens being decoded at once (fewer steps, higher efficiency), whereas larger values yield fewer tokens per step (more steps, better generation quality).

**Semi-Autoregressive Diffusion Decoding.** To further enhance quality and controllability, *Semi-Autoregressive Diffusion Decoding (SADD)* has been introduced. The idea is to divide the sequence into multiple *blocks* and generate them sequentially from left to right. Within each block, however, the MDM reverse process (with random or low-confidence remasking) is applied in parallel. This hybrid approach combines the global consistency of diffusion with the sequential structure of autoregression, yielding better performance on complex reasoning and dialogue tasks. This hybrid strategy has been employed in recent dLLMs such as LLaDA (Nie et al., 2025a) and MMaDA (Yang et al., 2025a).

## 2.2 PARALLEL DECODING ANALYSIS

DLLMs are designed to exploit their parallel decoding ability to accelerate the decoding process of LLMs, but most open-source dLLMs fall short of expectations because of their incompatibility between parallelism and accuracy. During decoding, the sampling procedure defined in Equation 4 produces only the marginal distribution for each token, $p(x_s^i \mid \mathbf{x}_t)$, for $i = \{1, \ldots, L\}$. However, parallel decoding requires access to the joint distribution over multiple tokens to be decoded simultaneously: $p(x_j^1, x_j^2, \ldots, x_j^k \mid \mathbf{X}_t)$, where $k$ denotes the number of tokens generated in one parallel decoding step. This discrepancy gives rise to a methodological challenge, namely that parallel decoding must approximate the joint distribution using only the available marginals $p(x_j^i \mid \mathbf{X}_t)$. Designing effective approximation strategies for bridging this gap constitutes a central problem in the development of parallel decoding algorithms.

To gain empirical insights into this theoretical inconsistency, we investigate how the positional distribution of previously decoded tokens affects the decoding process. In principle, the consistency of different decoding strategies can be quantitatively assessed by the Kullback–Leibler divergence $\mathrm{KL}(p_{\text{stepwise}}(\mathbf{x}) \,\|\, p_{\text{one-pass}}(\mathbf{x}))$. However, directly computing the standard KL divergence over multi-step generation is computationally intractable. Therefore, we employ an approximation where $n$ denotes the number of masked tokens, and their step-by-step generation is regarded as the ground truth. Specifically, given the logits $z^{\text{step}}$ from stepwise decoding, we take the most probable token index

$$i^* = \arg\max_{v \in \mathcal{V}} z_v^{\text{step}}, \tag{5}$$

and approximate the ground-truth distribution as one-hot, i.e., $p_{\text{stepwise}}(v) \approx \mathbf{1}_{[v=i^*]}$. For the one-pass prediction, we compute

$$p_{\text{one-pass}}(v) = \mathrm{softmax}(z^{\text{once}})_v. \tag{6}$$

Under this approximation, the KL divergence reduces to the negative log-likelihood of the one-pass model at $i^*$:

$$\mathrm{KL}_{\text{approx}} = -\sum_{j=1}^{n} \log p_{\text{one-pass}}(i_j^*), \tag{7}$$

where $i_j^*$ is the argmax token in the $j$-th masked position determined by stepwise decoding. This formulation can be interpreted as a cross-entropy surrogate of the KL divergence, where one-hot targets make the comparison tractable and are standard in language modeling (Goldberg, 2017).

We evaluate the proposed approximation on two representative benchmarks: HumanEval (Chen et al., 2021), which focuses on code generation, and GSM8k (Cobbe et al., 2021), which targets mathematical reasoning. The experimental results are presented in Figure 1. On both benchmarks, the approximate KL divergence under the consecutive masking strategy increases steadily as the number of masked tokens grows, with a markedly faster growth rate compared to the sparse masking strategy. In contrast, sparse masking maintains consistently low divergence across decoding steps. This observation suggests that sparse masking allows dLLMs to make better use of bidirectional self-attention. Specifically, by leaving unmasked anchor positions interleaved throughout the input, sparse masking enables the model to attend to reliable contextual signals from both left and right neighborhoods of each masked position, thereby improving the robustness and consistency of parallel decoding.

These results suggest that the sparse mask offers substantial advantages in mitigating distributional shift and maintaining decoding stability. This empirical robustness is consistent with our preliminary observation: when most tokens in a sequence are already decoded, the undecoded tokens approximate the ground truth more closely if they are sparsely scattered across the text rather than being continuously clustered. Motivated by this finding, we hypothesize that if undecoded tokens can be structurally organized to mimic such sparse distributions through an appropriate decoding strategy, it becomes possible to accelerate the generation process while preserving or even improving decoding accuracy.

## 3 METHODOLOGY

Building on our preliminary study, we find that sparse masking—where undecoded tokens remain sparsely scattered—suppresses distributional shift and stabilizes decoding. This effect arises be-

cause sparsity enables more effective use of bidirectional attention, guiding predictions toward the ground truth. Motivated by this observation, we propose a divide-and-conquer framework that partitions undecoded sequences into smaller subproblems, allowing parallel resolution that both accelerates generation and reduces bias.

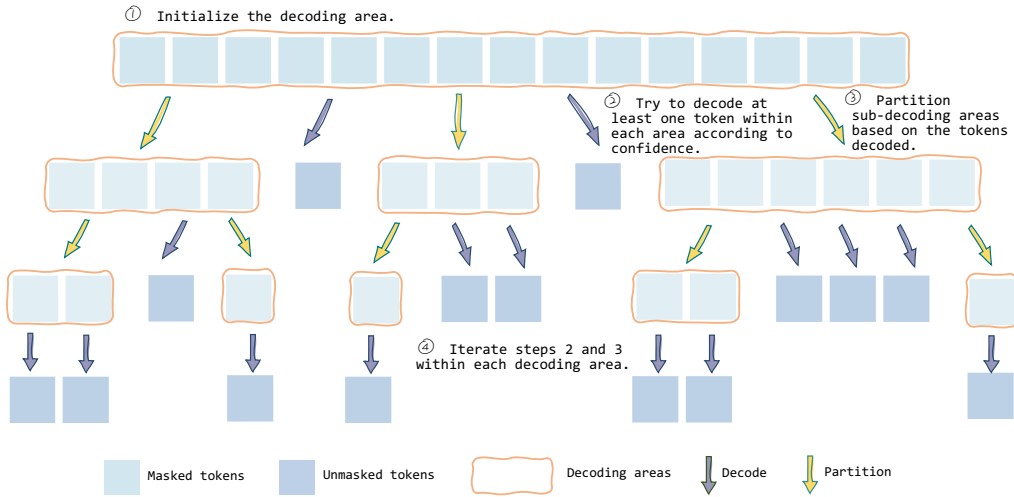

Figure 2: **Illustration of Hierarchy-dLLM**. The decoding process starts by initializing a decoding area, then decoding tokens based on confidence, and partitioning sub-areas according to the decoded tokens. Steps 2 and 3 are iteratively applied within each sub-decoding area, enabling efficient multi-token decoding while preserving accuracy.

## 3.1 DIVIDE-AND-CONQUER DECODING STRUCTURE

To achieve efficient and stable text generation, we design the model with a *divide-and-conquer decoding structure*, which progressively resolves masked spans through an iterative process of initialization, decoding, and subdivision. This design seeks to balance decoding efficiency with generation accuracy by breaking down the complex decoding task into smaller, well-structured units. The hierarchical organization prevents the model from predicting overly dependent tokens in the same step, converting the difficulty of parallel decoding into a series of more tractable sub-problems.

**Initialization stage**. Before decoding begins in each block, the block is represented as a contiguous span of masked tokens with a predefined length $l$. This masked span serves as the initial sub-decoding areas, providing a structured starting point for the generation process.

**Decoding stage**. Each sub-decoding area is processed independently and in parallel, following the decoding strategy introduced in Section 3.2. The objective is to maximize decoding efficiency while trying to ensure that each block yields at least one valid token. By decoding in parallel across multiple sub-areas, the model inherently possesses significant structural potential for acceleration.

**Subdivision stage**. Tokens generated in the previous step act as anchors to partition the remaining masked regions. Every contiguous span of undecoded tokens is split into smaller, independent sub-decoding areas, which are then processed in the next iteration. This recursive partitioning gradually reduces the decoding problem to smaller segments, simplifying the generation task.

The decoding and subdivision stages are repeated iteratively until no masked tokens remain in any decoding block. Through this iterative refinement process, the model incrementally resolves all masked positions while maintaining stability and coherence in the generated sequence. Overall, the divide-and-conquer decoding structure provides a principled framework that achieves $O(\log n)$-level acceleration under ideal conditions while fully exploiting the rich contextual information inherent in the bidirectional attention mechanism of dLLMs. This design not only ensures substantial efficiency gains but also preserves decoding accuracy, thereby offering a scalable and reliable foundation for subsequent stages of our model.

## 3.2 Decoding Strategies Within Sub-decoding Areas

The greatest challenge in the divide-and-conquer structure lies in how to decode effectively within each sub-decoding area. Our objective is to decode as many tokens as possible at each step while minimizing the risk of introducing errors that propagate through later iterations. To formalize this, let $p_\theta(x_s^i \mid X_t)$ denote the model's posterior probability of predicting token $x_s^i$ at position $i$ in the $s$-th decoding step, given the current corrupted sequence $X_t$. We then define the confidence score for position $i$ as

$$c^i = \max_{v \in \mathcal{V}} p_\theta(x_s^i = v \mid X_t). \tag{8}$$

A natural starting point is to decode tokens only when they are sufficiently reliable. Let $\mathcal{A}$ denote a sub-decoding area. Concretely, whenever $c^i$ surpasses a high threshold $\tau_{\text{high}}$,

$$x_s^i = \arg\max_{v \in \mathcal{V}} p_\theta(x_s^i = v \mid X_t), \quad \text{if} \quad c^i \geq \tau_{\text{high}}, \quad i \in \mathcal{A}, \tag{9}$$

the corresponding token is committed to the sequence. This simple rule favors semantic stability, since only high-confidence tokens are introduced, and it also allows multiple positions within $\mathcal{A}$ to be decoded in parallel when the evidence is strong.

Yet in practice, some sub-decoding areas may not contain any tokens above the high threshold, as the underlying probability distribution over the vocabulary can be relatively flat in these regions, leaving all candidate tokens with comparable but low confidence scores. If we decode nothing in such cases, progress slows dramatically; if we force a decision, performance can suffer. To mitigate this trade-off, we relax the condition: when no position meets equation 9, we still allow one token to be decoded, namely the most confident candidate in $\mathcal{A}$, provided that its confidence exceeds a lower threshold $\tau_{\text{low}}$,

$$x_s^{i^*} = \arg\max_{v \in \mathcal{V}} p_\theta(x_s^{i^*} = v \mid X_t), \quad \text{if} \quad i^* = \arg\max_{i \in \mathcal{A}} c^i, \quad c^{i^*} \geq \tau_{\text{low}}. \tag{10}$$

This adaptive rule ensures that each area contributes meaningfully while preventing the premature incorporation of extremely unreliable predictions.

These two conditions, however, may occasionally leave an iteration without any decoded tokens. This typically happens in more challenging cases, where the model cannot confidently commit to a prediction, so that the probability distribution over the vocabulary is relatively flat across positions and all candidate tokens fall below the relaxed threshold $\tau_{\text{low}}$. To avoid stalling, we enforce steady progress by always decoding the globally most confident position when necessary:

$$x_s^{i^\dagger} = \arg\max_{v \in \mathcal{V}} p_\theta(x_s^{i^\dagger} = v \mid X_t), \quad \text{if} \quad i^\dagger = \arg\max_{i \in \{1,\dots,L\}} c^i. \tag{11}$$

This fallback guarantees that every step yields at least one decoded token.

Finally, as decoding advances, early predictions can become inconsistent with the evolving context, reflected by a noticeable confidence drop. To adaptively correct such cases, we introduce a remasking step: before repartitioning into sub-decoding areas, all decoded tokens are checked, and any token with $c^i < \tau_{\text{remask}}$ is replaced by the mask symbol $[MASK]$,

$$x_s^i = [MASK] \quad \text{if} \quad c^i < \tau_{\text{remask}}. \tag{12}$$

This prevents error accumulation and helps maintain global coherence throughout the sequence.

Taken together, our decoding strategy begins with a strict high-threshold rule, then gradually relaxes through a controlled low-threshold selection, incorporates a fallback to guarantee steady progress, and finally applies a remasking step to revise unreliable predictions. In following this progressively relaxed procedure, the strategy adheres to a best-effort principle, since it encourages decoding whenever trustworthy evidence is available while postponing or correcting low-confidence tokens, thereby balancing efficiency, token-level reliability, and contextual coherence under the bidirectional attention mechanism of dLLMs.

# 4 Experiments

## 4.1 Experiment settings

We implement the proposed Hierarchy-dLLM framework on three open-source models: `llada-instruct-8B`, `llada-1.5-8B`, and `Dream-7B`, and evaluate it on four widely used

Table 1: **Main results of Hierarchy-dLLM on the LLaDA-Instruct-8B model and LLaDA-1.5-8B model across five benchmarks.** We report task performance (**Accuracy Score**) and decoding efficiency. Efficiency is measured by **TPS** (throughput per second), reflecting practical throughput, and **TPF** (tokens per forward call), indicating how many tokens are decoded per model invocation. Values in parentheses denote the relative **performance change** compared to the baseline and the **speedup factor** with respect to decoding efficiency. The best performance and highest TPS/TPF are highlighted in bold.

| Task | Method | LLaDA-Instruct-8B | | | LLaDA-1.5-8B | | |
| | | Performance | Speed | | Performance | Speed | |
| | | Score ↑ | TPF ↑ | TPS ↑ | Score ↑ | TPF ↑ | TPS ↑ |
|---|---|---|---|---|---|---|---|
| GSM8K | Vanilla | 77.26 | 0.52 | 2.35 | 83.17 | 0.65 | 1.99 |
| | Fast-dLLM | **77.86(0.6+)** | 2.85(5.48×) | 12.80(5.45×) | 83.32(0.15+) | 3.10(4.77×) | 9.54(4.79×) |
| | WINO | 68.75(8.51−) | 3.24(6.23×) | 10.95(4.66×) | 81.25(1.92−) | 3.97(6.10×) | 8.00(4.02×) |
| | Hierarchy-dLLM | 77.18(0.08−) | **3.79(7.29×)** | **19.41(8.26×)** | **83.70(0.53+)** | **4.25(6.54×)** | **14.83(7.45×)** |
| Math500 | Vanilla | 41.20 | 0.83 | 7.87 | 39.80 | 0.84 | 7.96 |
| | Fast-dLLM | 40.60(0.6−) | 2.71(3.27×) | 24.99(3.17×) | 39.40(0.4−) | 2.79(3.32×) | 25.81(3.24×) |
| | WINO | 37.40(3.8−) | 3.78(4.55×) | 30.73(3.90×) | 41.00(1.2+) | 3.75(4.46×) | 30.48(3.83×) |
| | Hierarchy-dLLM | **41.60(0.4+)** | **3.53(4.25×)** | **37.34(4.74×)** | **41.60(1.8+)** | **3.99(4.75×)** | **42.25(5.31×)** |
| HumanEval | Vanilla | 43.90 | 0.93 | 8.56 | 43.29 | 0.93 | 8.56 |
| | Fast-dLLM | 43.90(0+) | 2.94(3.16×) | 27.12(3.17×) | 42.07(1.22−) | 2.97(3.19×) | 27.45(3.21×) |
| | WINO | 43.90(0+) | 3.80(4.08×) | 28.15(3.29×) | 42.68(0.61−) | 3.84(4.12×) | 28.53(3.33×) |
| | Hierarchy-dLLM | **44.51(0.61+)** | **3.93(4.23×)** | **41.52(4.85×)** | **45.12(1.83+)** | **4.20(4.52×)** | **44.18(5.16×)** |
| MBPP | Vanilla | 37.60 | 0.13 | 0.65 | 40.20 | 0.16 | 0.80 |
| | Fast-dLLM | 37.60(0+) | 1.73(10.81×) | 7.26(11.17×) | **40.40(0.2+)** | 1.73(10.81×) | 8.40(10.5×) |
| | WINO | **38.40(0.8+)** | 1.27(9.76×) | 4.56(7.02×) | 34.80(5.4−) | 1.37(8.56×) | 4.92(6.15×) |
| | Hierarchy-dLLM | 37.60(0+) | **2.03(15.62×)** | **11.20(17.23×)** | **40.40(0.2+)** | **2.29(14.31×)** | **12.70(15.88×)** |
| IF-Eval | Vanilla | **57.67** | 0.59 | 6.28 | 58.78 | 0.59 | 6.29 |
| | Fast-dLLM | 57.30(0.37−) | 1.45(2.46×) | 15.46(2.46×) | 58.23(0.55+) | 1.38(2.34×) | 14.43(2.29×) |
| | WINO | 55.64(2.03−) | 1.64(2.78×) | 14.33(2.28×) | 58.60(0.18−) | 1.59(2.69×) | 13.95(2.22×) |
| | Hierarchy-dLLM | 57.12(0.45−) | **1.82(3.08×)** | **22.14(3.53×)** | **59.15(0.37+)** | **1.74(2.95×)** | **21.21(3.37×)** |

benchmarks: GSM8K and MATH500 (Lightman et al., 2023) for mathematical reasoning, HumanEval and MBPP (Austin et al., 2021) for code generation, and IF-Eval (Zhou et al., 2023) for instruction following, with few-shot settings adopted in accordance with Nie et al. (2025b) and Zhu et al. (2025). To provide a comprehensive assessment of performance and efficiency, we compare Hierarchy-dLLM against vanilla autoregressive decoding, the parallel decoding scheme of Fast-dLLM (Wu et al., 2025), and a more complex multi-stage decoding method of WINO (Hong et al., 2025). All experiments are run on a single NVIDIA H20 GPU. Unless otherwise specified, the block size is set to 32 and the generation length to 512. For hyperparameter tuning, we conduct a grid search where $\tau_{high}$ ranges from 0.78 to 0.88, $\tau_{low}$ ranges from 0.3 to 0.5, and $\tau_{remask}$ is chosen in 0.01, 0.3 and 0.35. The exact settings and implementation details are available in our released code. We report performance using Pass@1 accuracy, and efficiency is measured with tokens per forward call (TPF) and throughput per second (TPS). Note that TPS excludes the `eos` token, and for consistency, we also exclude the `eos` token in TPF. All evaluations are conducted with the `lm-eval` (Gao et al., 2024) library to ensure consistency and reproducibility.

## 4.2 MAIN RESULTS

Across five benchmarks and three models, Hierarchy-dLLM consistently delivers strong accuracy while achieving the highest decoding efficiency.

Based on the experimental results on LLaDA-1.5-8B and LLaDA-Instruct-8B in 1, Hierarchy-dLLM achieves the best of both worlds—higher accuracy than both vanilla and Fast-dLLM baselines while providing the fastest decoding. The method shows the most notable gains on mathematical reasoning tasks of GSM8K and Math500, where it not only accelerates throughput by up to 17× over baselines but also improves accuracy by about 1 point, indicating its ability to mitigate error accumulation in long reasoning chains. On code generation tasks of HumanEval and MBPP, Hierarchy-dLLM

maintains or slightly improves accuracy while substantially increasing speed, with TPF gains up to $10\times$, underscoring its suitability for deterministic token generation.

On the Dream-7B model(see Table 2), we observe that Hierarchy-dLLM still brings comparable speedup gains, achieving large improvements in both TPF and TPS across all four benchmarks while maintaining stable performance. This confirms that the hierarchical mechanism consistently enhances decoding efficiency even on models trained with different architectures. However, compared to LLaDA, the absolute performance of Dream-7B with Hierarchy-dLLM achieves comparable speedups while its accuracy drop is no larger than, and sometimes smaller than, Fast-dLLM. We attribute this to Dream originating from an autoregressive base model, which provides weaker inherent support for parallel decoding and thus limits quality preservation compared to models with stronger parallelism.

Overall, Hierarchy-dLLM offers a unified acceleration framework that simultaneously improves or preserves task performance while substantially reducing inference cost across diverse settings.

Table 2: **Main results of Hierarchy-dLLM on the Dream-7B model across four benchmarks.**

| Task | Method | Performance | Speed | |
|---|---|---|---|---|
| | | Score ↑ | TPF ↑ | TPS ↑ |
| GSM8K | Vanilla | **75.8** | 1.00 | 4.76 |
| | Fast-dLLM | 73.8(2−) | 2.08(2.08×) | 9.30(1.95×) |
| | Hierarchy-dLLM | 73.2(2.6−) | **2.61(2.61×)** | **12.39(2.60×)** |
| Math500 | Vanilla | **17.8** | 1.00 | 10.41 |
| | Fast-dLLM | 12.0(5.8−) | 2.32(2.32×) | 21.45(2.06×) |
| | Hierarchy-dLLM | 16.0(1.8−) | **3.17(3.17×)** | **32.54(3.13×)** |
| HumanEval | Vanilla | **54.9** | 1.00 | 9.69 |
| | Fast-dLLM | 50.6(4.3−) | 2.08(2.08×) | 18.12(1.87×) |
| | Hierarchy-dLLM | 51.8(3.1−) | **2.61(2.61×)** | **25.32(2.61×)** |
| MBPP | Vanilla | **56.8** | 1.0 | 5.97 |
| | Fast-dLLM | 54.6(2.2−) | 4.35(4.35×) | 23.57(3.96×) |
| | Hierarchy-dLLM | 54.4(2.4−) | **5.06(5.06×)** | **29.89(5.01×)** |

## 4.3 ABLATION STUDY AND ANALYSIS

Unless otherwise specified, all ablation studies are conducted on the GSM8k dataset using the LLaDA-1.5-8B model. The generation length is fixed to 512 and the block length to 32, with all other hyperparameters kept identical to those in Section 4.1.

**Impact of different Generation Length.** To investigate the impact of block length, we fix the generation length to 512 and evaluate the performance of the baseline model and Hierarchy-dLLM with block sizes of 16, 32, and 64. As shown in Table 3, while the performance of both methods remains stable across different block sizes, Hierarchy-dLLM consistently achieves significantly higher TPF and TPS compared to vanilla decoding. Moreover, its advantage becomes more pronounced as the block length increases, demonstrating that the hierarchical design effectively sustains high throughput under larger decoding blocks.

**Impact of different Block Length.** As reported in Table 4, the TPS of dLLMs decreases as the generation length grows, which can be attributed to the bidirectional self-attention mechanism: longer sequences require more computation per decoding step. Although Hierarchy-dLLM is also affected, the slowdown is considerably mitigated compared to vanilla decoding. Consequently, the speedup ratio of Hierarchy-dLLM increases with generation length, while its performance exhibits a similar upward trend to the baseline, demonstrating stable efficiency and effectiveness under longer decoding scenarios.

**Effects of adjusting the threshold and low-threshold hyperparameters.** The effect of threshold and low threshold settings on performance and speed is examined when remasking is disabled. Figure 3a reports the score and TPS as the high threshold $\tau_{\text{high}}$ varies, with low threshold $\tau_{\text{low}}$ fixed at 0.3. Figure 3b shows the counterpart results when $\tau_{\text{low}}$ varies with $\tau_{\text{high}}$ fixed at 0.82. Across both

Table 3: **Performance and Speed on GSM8K with Different Generation Lengths.**

| Gen Length | Method | Performance | Speed | |
|---|---|---|---|---|
| | | Score ↑ | TPF ↑ | TPS ↑ |
| 256 | LLaDA-1.5-8B | 82.29 | 0.97 | 4.13 |
| | Hierarchy-dLLM | 81.34 | 4.38 (4.52×) | 18.53 (4.49×) |
| 512 | LLaDA-1.5-8B | 83.17 | 0.65 | 1.99 |
| | Hierarchy-dLLM | 83.70 | 4.25 (6.54×) | 14.83 (6.45×) |
| 1024 | LLaDA-1.5-8B | 84.38 | 0.26 | 0.76 |
| | Hierarchy-dLLM | 84.31 | 3.09 (11.88×) | 9.06 (11.92×) |

Table 4: **Performance and Speed on GSM8K with Different Block Lengths**

| Block Length | Method | Performance | Speed | |
|---|---|---|---|---|
| | | Score ↑ | TPF ↑ | TPS ↑ |
| 16 | LLaDA-1.5-8B | 83.40 | 0.69 | 2.22 |
| | Hierarchy-dLLM | 81.58 | 3.52 (5.10×) | 12.89 (5.81×) |
| 32 | LLaDA-1.5-8B | 83.17 | 0.66 | 2.13 |
| | Hierarchy-dLLM | 83.70 | 4.25 (6.44×) | 14.83 (6.96×) |
| 64 | LLaDA-1.5-8B | 83.85 | 0.64 | 1.98 |
| | Hierarchy-dLLM | 81.35 | 4.69 (7.23×) | 17.18 (8.68×) |

settings, the performance of Hierarchy-dLLM remains relatively stable at a high level, indicating robustness to threshold choices. By contrast, TPS is more sensitive, which means increasing $\tau_{\text{high}}$ leads to a noticeable decline in efficiency, while changes in $\tau_{\text{low}}$ only cause minor variations in TPS. These findings suggest that selecting a moderately small $\tau_{\text{high}}$ is crucial to balancing accuracy and efficiency, whereas $\tau_{\text{low}}$ has a negligible impact, reflecting that the model is resilient to uncertainty pruning in low-confidence regions.

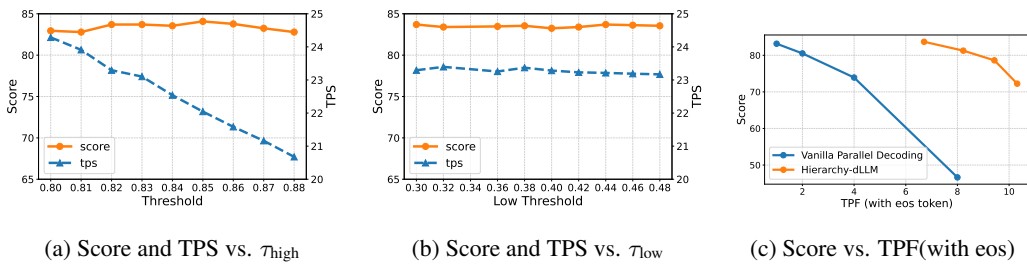

(a) Score and TPS vs. $\tau_{\text{high}}$     (b) Score and TPS vs. $\tau_{\text{low}}$     (c) Score vs. TPF(with eos)

Figure 3: **Joint Analysis of Accuracy and Efficiency On GSM8K** (a) Score and TPS with varying $\tau_{\text{high}}$ while fixing $\tau_{\text{low}} = 0.3$. (b) Score and TPS with varying $\tau_{\text{low}}$ while fixing $\tau_{\text{high}} = 0.82$. (c) Comparison of score versus TPF between vanilla parallel decoding and Hierarchy-dLLM.

**Comparison with naive parallel sampling.** We further compare Hierarchy-dLLM with a vanilla parallel decoding strategy where a fixed number of tokens, including the `eos` token, are sampled in each step so that the total token count divided by sampling steps matches the intended parallel factor. As shown in Figure 3c, increasing the number of tokens per step rapidly degrades the performance of vanilla parallel decoding despite the speed gain, leading to a poor speed–accuracy trade-off. In contrast, Hierarchy-dLLM maintains consistently high performance even under high TPF, while preserving substantial speed improvements. This demonstrates that Hierarchy-dLLM achieves a more favorable balance between efficiency and accuracy compared to naive parallel decoding.

## 5 RELATED WORKS

### 5.1 DIFFUSION LARGE LANGUAGE MODELS

Diffusion modeling for language has recently gained significant attention. Early work extended score matching to discrete spaces with score entropy to build Score Entropy Discrete Diffusion models (Lou et al., 2024). To simplify and generalize this paradigm, masked diffusion formulations reinterpreted objectives as weighted cross-entropy losses, leading to improved training and flexibility (Shi et al., 2025). Scaling studies further demonstrated that Masked Diffusion Models follow comparable scaling laws to autoregressive methods and enable efficient compute–quality trade-offs (Nie et al., 2025a).

At larger scale, diffusion has been positioned as a credible foundation model alternative: LLaDA shows diffusion-based LLMs trained with pretraining and fine-tuning can rival strong AR models like LLaMA and even surpass GPT-4o on specific tasks (Nie et al., 2025b); follow-up work LLaDA2.0 (Bie et al., 2025) further scales this line via AR-to-dLLM conversion up to 100B parameters with block-level diffusion training, and LLaDA2.1 (Bie et al., 2026) improves the speed–quality

trade-off by combining Mask-to-Token diffusion with Token-to-Token editing and RL-based alignment for dLLMs.

Extensions to robotics and multimodality include LLaDA-VLA (Wen et al., 2025), which adapts diffusion VLMs for vision-language-action policy learning, and MMaDA (Yang et al., 2025a), which introduces a unified multimodal diffusion architecture with shared reasoning and generation abilities. Complementary to the above, DREAM (HKUNLP, 2025) explores discrete reasoning in autoregressive models to improve controllability and reasoning quality.

## 5.2 ACCELERATION TECHNIQUES FOR DLLMS

Recent work on efficient inference for dLLMs has rapidly expanded along two main axes: (i) parallel decoding strategies, and (ii) caching techniques that reduce the cost of bidirectional attention and KV storage. A number of system-level frameworks further integrate these algorithmic advances into unified inference stacks.

**Parallel Decoding Strategies**  To accelerate dLLM inference, researchers leverage token confidence. Fast-dLLM (Wu et al., 2025) proposes a training-free, confidence-aware parallel decoding for dLLMs. CreditDecoding (Wang et al., 2025) utilizes "Trace Credits" based on historical logits to accelerate the convergence of underconfident tokens. LocalLeap (Kong et al., 2025) employs local determinism propagation around high-confidence anchors to reduce decoding steps. Dimple (Yu et al., 2025b) applies a confident decoding strategy dynamically adjusting token generation counts to significantly reduce iterations. Several approaches introduce verification steps to ensure quality while accelerating inference. WINO (Hong et al., 2025) introduces a revocable decoding mechanism that aggressively drafts tokens and uses bidirectional context to re-mask and refine errors. Adaptive Parallel Decoding (Israel et al., 2025) inverts standard speculation by mixing dLLM marginals with a small autoregressive model to dynamically adjust sample sizes. Other works focus on sampling dynamics and model structure. SlowFast Sampling (Wei et al., 2025) alternates between exploratory and accelerated decoding stages based on certainty and convergence principles. Addressing fixed-length limitations, dLLM-Var (Yang et al., 2025b) enables native variable generation lengths by training the model to predict [EOS] tokens.

**KV Cache Optimization**  Given the bidirectional attention costs in dLLMs, efficient cache management is critical. Fast-dLLM (Wu et al., 2025) introduces a block-wise approximate KV cache tailored for bidirectional models, enabling cache reuse with negligible performance drop alongside confidence-aware decoding. DLLM-Cache (Liu et al., 2025) proposes a training-free framework that combines long-interval prompt caching with partial response updates guided by feature similarity. To address memory constraints, Sparse-dLLM (Song et al., 2025a) leverages persistent cross-layer sparsity to dynamically evict unimportant tokens while retaining pivotal ones, maintaining performance with reduced memory usage. $D^2$Cache (Jiang et al., 2025) reuses KV states for stable tokens via a dual adaptive cache. MaskKV (Huang et al., 2025) leverages mask-token–guided scoring and adaptive budgeting for fine-grained KV compression.

## 6 CONCLUSION

In this work, we introduced Hierarchy-dLLM, a hierarchical decoding framework that applies the divide-and-conquer principle to accelerate large language model inference. By recursively partitioning masked spans into smaller sub-decoding areas and decoding tokens according to confidence, our method effectively increases the number of tokens generated per step, thereby improving information utilization. Experiments on multiple benchmarks show that Hierarchy-dLLM maintains comparable or even better accuracy than existing approaches, while achieving up to $17\times$ speedup over vanilla decoding and about $1.5\times$ faster than Fast-dLLM. These results demonstrate that hierarchical, divide-and-conquer decoding provides a practical and scalable solution for efficient diffusion-based generation. While our current framework is entirely *training-free*, an exciting future direction is to perform post-training adaptations so that the model distribution better aligns with hierarchical decoding, potentially further enhancing both efficiency and effectiveness.

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

## A  LLM Usage Statement

In accordance with the ICLR policy on Large Language Model (LLM) usage, we disclose that LLMs (e.g., ChatGPT) were used solely as a writing assistant for language polishing and minor text refinement including improving readability, grammar, and style. The model was not involved in research ideation, experimental design, data collection, analysis, or interpretation of results. All scientific contributions and substantive content were generated entirely by the authors.

## B  Pseudocode of Hierarchy-dLLM

For clarity and reproducibility, we provide the pseudocode of the proposed Hierarchy-dLLM decoding algorithm in Algorithm 1. Equation 13 summarizes the decoding rule across all sub-decoding areas. The strategy follows a progressive relaxation principle: (i) tokens above the high threshold are committed directly; (ii) if no such tokens exist, we fall back to the most confident position within each area above the low threshold; (iii) if still no commitment is possible, the model globally commits to the most confident token to guarantee progress; and (iv) after each step, all committed tokens are re-evaluated, and low-confidence ones are remasked. This unified rule concisely encodes the decision-making logic underlying Algorithm 1 in the pseudocode, ensuring both efficiency and robustness during divide-and-conquer decoding.

$$x_s^i = \begin{cases} \arg\max_{v \in \mathcal{V}} p_\theta(x_s^i = v \mid X_t), & \text{if } c^i \geq \tau_{\text{high}}, \ i \in \mathcal{A} \\ \arg\max_{v \in \mathcal{V}} p_\theta(x_s^{i^*} = v \mid X_t), & \text{if } i^* = \arg\max_{j \in \mathcal{A}} c^j, \ c^{i^*} \geq \tau_{\text{low}} \\ \arg\max_{v \in \mathcal{V}} p_\theta(x_s^{i^\dagger} = v \mid X_t), & \text{if } i^\dagger = \arg\max_{j \in \{1,\dots,L\}} c^j \\ \texttt{[MASK]}, & \text{otherwise} \end{cases} \quad (13)$$

---

**Algorithm 1:** Hierarchy-dLLM

---

**Input:** Prompt: $p$; block length: $L$; number of blocks: $N$; thresholds: $\tau_{\text{low}}, \tau_{\text{high}}, \tau_{\text{remask}}$
**Output:** Decoded sequence
Initialize sequence $x \leftarrow p \,\|\, [MASK]^{L \times N}$;
// $p$ concatenated with $L \times N$ masks
**for** $i \leftarrow 1$ **to** $N$ **do**
  // Select the $i$-th block of $L$ masks
  Let $B \leftarrow$ positions of masks $[(i-1)L+1, iL]$ in $x$;
  **while** *not all tokens in B decoded* **do**
    // Remask step (only within block $B$)
    **foreach** *token $x_j \in B$* **do**
      **if** $conf(x_j) < \tau_{remask}$ **then**
        $\lfloor$ set $x_j \leftarrow [MASK]$;

    Divide block $B$ into sub-decoding areas;
    $\mathcal{D} \leftarrow \emptyset$;
    // decoded tokens in this iteration
    **foreach** *sub-decoding area $A \subseteq B$* **do**
      $t^* \leftarrow \arg\max_{t \in A} conf(t)$;
      $c^* \leftarrow conf(t^*)$;
      **if** $c^* \geq \tau_{high}$ **then**
        $\lfloor$ decode $t^*$; add to $\mathcal{D}$;
      **if** $t^*$ *is maximal in A* **and** $c^* \geq \tau_{low}$ **then**
        $\lfloor$ decode $t^*$; add to $\mathcal{D}$;

    **if** $\mathcal{D} = \emptyset$ **then**
      $\lfloor$ decode token with highest confidence across the entire block $B$;

**return** decoded sequence;

---

## C    IMPACT OF DIFFERENT GENERATION LENGTH ON OTHER TASKS

We further examine the impact of generation length by evaluating Math500, HumanEval, and MBPP under lengths of 256, 512, and 1024 tokens. As reported in Table 5 the task accuracy of both vanilla decoding and Hierarchy-dLLM remains stable across different generation lengths, indicating that extending sequence length does not harm the correctness of generated outputs. In contrast, the efficiency metrics reveal a clear distinction: Hierarchy-dLLM consistently yields substantially higher TPF and TPS than vanilla decoding across all datasets. This advantage is especially pronounced at longer generation lengths, where vanilla decoding exhibits severe throughput degradation while Hierarchy-dLLM maintains high sampling efficiency. These findings confirm that the hierarchical design not only sustains accuracy but also scales favorably with longer contexts, making it particularly advantageous for tasks requiring extended generations.

## D    IMPACT OF DIFFERENT BLOCK LENGTH ON OTHER TASKS

To better understand the role of block length in hierarchical decoding, we evaluate Math500, HumanEval, and MBPP with block sizes of 16, 32, and 64. As shown in Table 6, task-level performance

Table 5: **Performance and Speed on Other Tasks with Different Generation Lengths.**

| Gen Length | Task | Method | Performance | Speed | |
|---|---|---|---|---|---|
| | | | Score ↑ | TPF ↑ | TPS ↑ |
| 256 | Math500 | Vanilla | 35.8 | 0.98 | 16.07 |
| | | Hierarchy-dLLM | 33.40 | 3.56 (3.68×) | 59.18 (3.68×) |
| | HumanEval | Vanilla | 42.68 | 0.97 | 14.01 |
| | | Hierarchy-dLLM | 35.98 | 4.45 (4.59×) | 54.50 (3.89×) |
| | MBPP | Vanilla | 40.8 | 0.34 | 2.12 |
| | | Hierarchy-dLLM | 39.6 | 2.91 (8.56×) | 19.81 (9.34×) |
| 512 | Math500 | Vanilla | 39.80 | 0.84 | 7.96 |
| | | Hierarchy-dLLM | 41.60 | 3.99 (4.75×) | 42.25 (5.31×) |
| | HumanEval | Vanilla | 43.29 | 0.93 | 8.56 |
| | | Hierarchy-dLLM | 45.12 | 4.20 (4.52×) | 44.18 (5.16×) |
| | MBPP | Vanilla | 40.40 | 0.16 | 0.80 |
| | | Hierarchy-dLLM | 40.40 | 2.29 (14.31×) | 12.70 (15.88×) |
| 1024 | Math500 | Vanilla | 42.2 | 0.63 | 3.62 |
| | | Hierarchy-dLLM | 40.20 | 4.43 (7.03×) | 28.43 (7.85×) |
| | HumanEval | Vanilla | 43.90 | 0.54 | 2.92 |
| | | Hierarchy-dLLM | 43.90 | 3.70 (6.85×) | 22.42 (7.68×) |
| | MBPP | Vanilla | 40.60 | 0.06 | 0.22 |
| | | Hierarchy-dLLM | 39.00 | 1.55(25.83×) | 6.17(28.5×) |

remains broadly comparable between vanilla decoding and Hierarchy-dLLM, with only minor fluctuations when block sizes increase from 16 to 64. This suggests that the hierarchical decoding strategy does not compromise output correctness even when operating under different structural granularities.

In terms of efficiency, however, the differences are striking. For all three datasets, Hierarchy-dLLM consistently delivers much higher TPF and TPS values, often exceeding vanilla decoding by an order of magnitude. At small block sizes of 16, throughput already improves significantly, with Hierarchy-dLLM showing 4×–10× improvements across tasks. When block size grows to 32 and 64, the advantage becomes even more pronounced: in HumanEval and MBPP, Hierarchy-dLLM exhibits more than 15× speedup relative to vanilla TPS, underscoring its ability to maintain high parallelism within and across sub-decoding areas.

## E  CASE STUDY

To illustrate Hierarchy-dLLM's decoding process, we present qualitative case studies on both GSM8K (Figure 4) and HumanEval (Figure 5). The GSM8K case with $\tau_{high} = 0.9$, $\tau_{low} = 0.4$, and remask disabled, visualizes hierarchical decoding with color-coded token steps, confirming that the model generates multiple tokens per step while reliably committing at least one token in each sub-decoding area. This ensures that reasoning chains are preserved without introducing inconsistency. The HumanEval case highlights how the same mechanism enables efficient multi-token generation in programming tasks, producing syntactically consistent partial code blocks across steps. These visualizations further reveal that decoding proceeds in parallel across sub-decoding areas, while within each area the model can also efficiently sample multiple tokens in parallel, thereby achieving both hierarchical structure and intra-step efficiency.

Importantly, these examples demonstrate that Hierarchy-dLLM strikes a desirable balance between quality and speed. On GSM8K, the model can generate correct intermediate steps and final answers, while still benefiting from faster decoding compared with vanilla left-to-right generation. On HumanEval, the model preserves program correctness and syntax while accelerating generation through parallel sampling. This combination of stable accuracy with substantial throughput gains reflects the core advantage of hierarchical decoding, as it avoids the typical trade-off between effectiveness and

Table 6: **Performance and Speed on Other Tasks with Different Block Lengths.**

| Block Length | Task | Method | Performance | Speed | |
|---|---|---|---|---|---|
| | | | Score ↑ | TPF ↑ | TPS ↑ |
| 16 | Math500 | Vanilla | 40.40 | 0.87 | 8.86 |
| | | Hierarchy-dLLM | 38.00 | 3.70 (4.25×) | 40.61 (4.58×) |
| | HumanEval | Vanilla | 42.07 | 0.94 | 8.61 |
| | | Hierarchy-dLLM | 42.68 | 3.69 (3.93×) | 38.56 (4.48×) |
| | MBPP | Vanilla | 41.20 | 0.17 | 0.86 |
| | | Hierarchy-dLLM | 40.00 | 1.66 (9.76×) | 8.87 (10.31×) |
| 32 | Math500 | Vanilla | 39.80 | 0.84 | 7.96 |
| | | Hierarchy-dLLM | 41.60 | 3.99 (4.75×) | 42.25 (5.31×) |
| | HumanEval | Vanilla | 43.29 | 0.93 | 8.56 |
| | | Hierarchy-dLLM | 45.12 | 4.20 (4.52×) | 44.18 (5.16×) |
| | MBPP | Vanilla | 40.40 | 0.16 | 0.80 |
| | | Hierarchy-dLLM | 40.40 | 2.29 (14.31×) | 12.70 (15.88×) |
| 64 | Math500 | Vanilla | 39.40 | 0.88 | 8.91 |
| | | Hierarchy-dLLM | 37.2 | 4.47 (5.08×) | 50.75 (5.70×) |
| | HumanEval | Vanilla | 40.85 | 0.93 | 8.73 |
| | | Hierarchy-dLLM | 37.80 | 4.24 (4.56×) | 44.61 (5.11×) |
| | MBPP | Vanilla | 34.8 | 0.13 | 0.63 |
| | | Hierarchy-dLLM | 34.20 | 2.23 (17.15×) | 12.30 (19.52×) |

efficiency and offers a unified approach applicable across reasoning and code tasks. The complete prompts used in both experiments are provided in the appendix for reproducibility.

---

**Prompt**

**Question:** Jen and Tyler are gymnasts practicing flips. Jen is practicing the triple-flip while Tyler is practicing the double-flip. Jen did sixteen triple-flips during practice. Tyler flipped in the air half the number of times Jen did. How many double-flips did Tyler do?
**Answer:** Jen did 16 triple-flips, so she did 16 * 3 = 48 flips. Tyler did half the number of flips, so he did $48/2 = 24$ flips. A double flip has two flips, so Tyler did $24/2 = 12$ double-flips.
#### 12

**Question:** Four people in a law firm are planning a party. Mary will buy a platter of pasta for $20 and a loaf of bread for $2. Elle and Andrea will split the cost for buying 4 cans of soda which cost $1.50 each, and chicken wings for $10. Joe will buy a cake that costs $5. How much more will Mary spend than the rest of the firm put together?
**Answer:** Mary will spend $22. Elle and Andrea together will spend $16, and with Joe's $5 the total is $21. So Mary spends $1 more.
#### 1

**Question:** A charcoal grill burns fifteen coals to ash every twenty minutes of grilling. The grill ran for long enough to burn three bags of coals. Each bag of coal contains 60 coals. How long did the grill run?
**Answer:** The grill burned $3 \times 60 = 180$ coals. Since 15 coals burn every 20 minutes, the grill ran for $(180/15) \times 20 = 240$ minutes.
#### 240

**Question:** A bear is preparing to hibernate... How many pounds did it gain eating small animals?
**Answer:** The bear gained 200 pounds from berries, 400 from acorns, 200 from salmon.

---

Remaining 200 are from small animals.
#### 200

**Question:** Brendan can cut 8 yards of grass per day... How many yards will Brendan be able to cut after a week?
**Answer:** With 50% more efficiency he cuts 12 yards/day. In 7 days: $12 \times 7 = 84$.
#### 84

**Question:** Skyler has 100 hats on his hand with the colors red, blue, and white. Half of the hats are red, 3/5 of the remaining hats are blue, and the rest are white. How many white hats does Skyler have?
**Answer:**

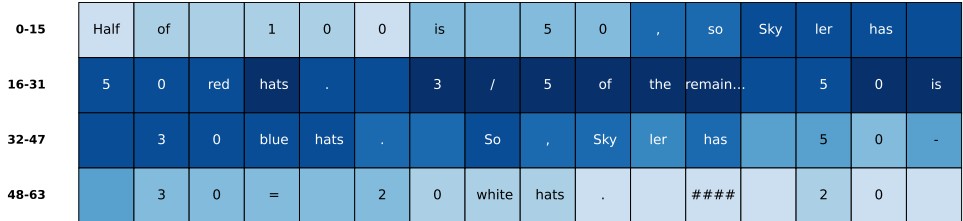

Figure 4: **Visualization of Decoding Steps on GSM8K**. Tokens are color-coded by their decoding step (lighter = earlier, darker = later). Thresholds are set to $\tau_{high} = 0.9$, $\tau_{low} = 0.4$, with remasking disabled. The figure illustrates that multiple tokens are generated in a single step, while each sub-decoding area reliably commits at least one token, consistent with the intended hierarchical decoding behavior.

**Prompt**

**Question:**
def compare(game,guess): ”””I think we all remember that feeling when the result of some long-awaited event is finally known. The feelings and thoughts you have at that moment are definitely worth noting down and comparing. Your task is to determine if a person correctly guessed the results of a number of matches.
You are given two arrays of scores and guesses of equal length, where each index shows a match.
Return an array of the same length denoting how far off each guess was. If they have guessed correctly, the value is 0, and if not, the value is the absolute difference between the guess and the score.
example:
compare([1,2,3,4,5,1],[1,2,3,4,2,-2]) - [0,0,0,0,3,3]
compare([0,5,0,0,0,4],[4,1,1,0,0,-2]) - [4,4,1,0,0,6] ”””

**Answer:**

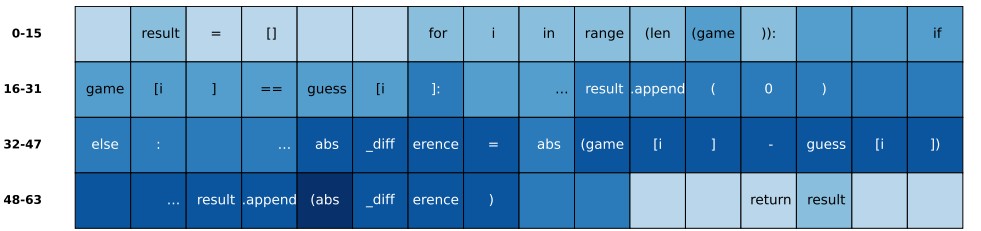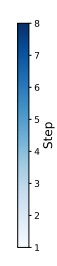

Figure 5: **Visualization of Decoding Steps on HumanEval**.

## F    ANOTHER INSIGHT ON SPARSE MASK AND CONSECUTIVE MASK

Building on the pre-study in Section 2, we further refine the KL analysis by replacing the multi-step model outputs with the original input text as the ground-truth target, and masking a larger number of tokens in one step. We then recompute the approximate KL divergence between the model's one-step predictions and this ground truth under both masking strategies. As shown in Figure 6, Sparse Mask consistently yields lower KL divergence than Consecutive Mask on both HumanEval and GSM8K, even when many tokens are masked simultaneously. These results indicate that scattered masking better preserves the semantic structure of the input text and is more suitable for accurate multi-token one-step generation than masking a contiguous block of tokens.

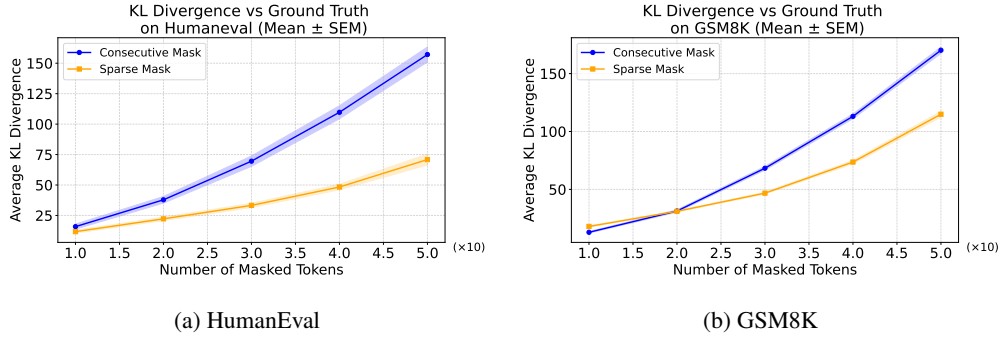

(a) HumanEval                                                 (b) GSM8K

Figure 6: **KL Divergence vs Ground Truth**

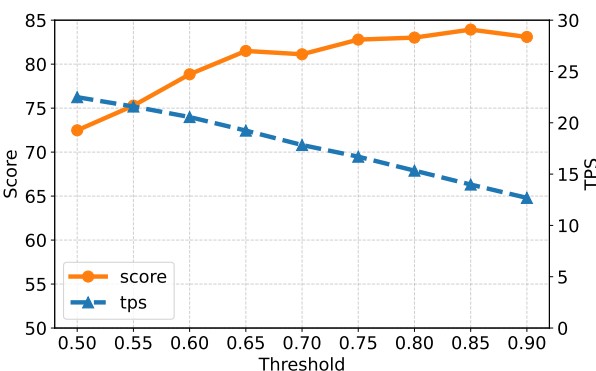

Figure 7: **Score and TPS vs. $\tau_{\text{high}}$ in a wide range**

## G    EXTENDED ANALYSIS OF THRESHOLD SENSITIVITY

Figure 7 shows the resulting task score and tokens-per-second (TPS) as functions of the threshold.

As expected, TPS decreases approximately linearly as the threshold grows: higher thresholds make the algorithm more conservative and reduce parallelism. Nonetheless, even at the highest threshold , the decoding speed is still significantly higher than standard autoregressive decoding (see in 1). There is a relatively flat region around thresholds $0.70$–$0.85$ where the score is near its optimum and TPS remains clearly above baseline. This suggests that in practice our method is not sensitive to the exact choice of the threshold: any value in this band yields a favorable balance between quality and speed.

## H    INFERENCE TIME OVERHEAD ANALYSIS

We implement Hierarchy-dLLM using efficient PyTorch tensor operations, so most computations run entirely on GPU without incurring heavy CPU-side control overhead. To quantify the runtime cost of our hierarchical selection logic, we benchmark both vanilla diffusion decoding and Hierarchy-dLLM on HumanEval using a single H20 GPU. As shown in 8, for LLaDA-Instruct-8B the token-selection and index-manipulation procedures in Hierarchy-dLLM account for only 1.2% of the end-to-end inference time, an increase of merely 0.5% compared to vanilla decoding, while the model forward pass remains the dominant cost. At the same time, Hierarchy-dLLM reduces the average number of function evaluations (NFE) from 128 to 64 and nearly halves the total decoding latency. This confirms that our method introduces negligible overhead beyond model forwards and can efficiently accelerate dLLM inference in practice.

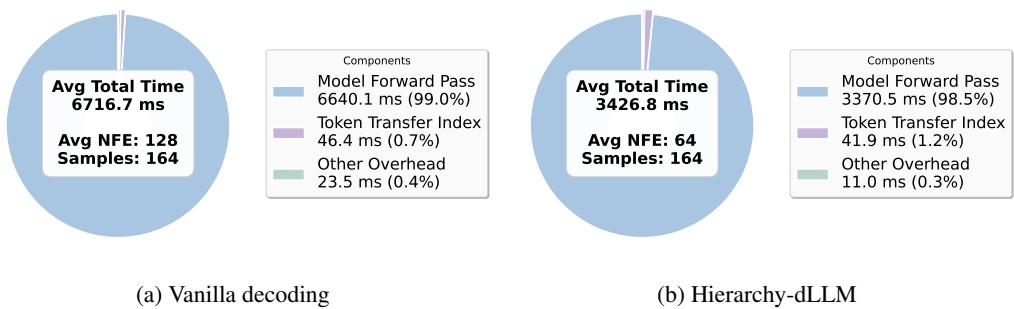

(a) Vanilla decoding                    (b) Hierarchy-dLLM

Figure 8: **Inference Time Analysis of LLada-Instruct-8B on HumanEval**

