# OpenReview forum: "Hierarchy Decoding: A Training-free Parallel Decoding Strategy  for Diffusion Large Language Models"
_ICLR.cc/2026/Conference — ICLR 2026 Poster_

### Official Review · Reviewer_G5aG · 2025-10-27

**Soundness:** 3
**Presentation:** 3
**Contribution:** 3
**Rating:** 6
**Confidence:** 2

**Summary:**

This paper proposes Hierarchy Decoding, a training-free divide-and-conquer decoding method for diffusion LLMs. It maintains sparse masking by iteratively committing high-confidence tokens and partitioning sequences into sub-decoding areas, reducing distributional drift. Experiments on reasoning and code tasks show 4–17× speed-ups with comparable or better accuracy, demonstrating an efficient and plug-and-play inference strategy for diffusion-based LLMs.

**Strengths:**

1.Proposes a training-free hierarchical decoding strategy that is conceptually simple yet original for diffusion LLMs.

2.Provides clear empirical motivation showing sparse masks yield more stable distributions than consecutive ones.

3.Achieves 5–17× decoding speed-ups with little or no accuracy loss across reasoning and code tasks.

**Weaknesses:**

1.The claimed O(log n) acceleration lacks formal assumptions and a proof or bound, so the theory feels anecdotal.

2.The KL analysis hinges on argmax→one-hot “ground truth,” which may not generalize beyond code/math settings.

3.Minor citation issue Line146: the text still shows “MMaDA (?)” and should be fixed to the actual reference.

**Questions:**

1.Does the sparse-mask advantage persist with non-argmax targets on open-ended tasks?

---

> ### Author Response · Authors · 2025-11-27
> **Response to Reviewer 4: Page 1**
>
> # Response to Reviewer 4
>
> We are very grateful for your positive assessment of our work and for the constructive suggestions you offered. Your comments helped us identify several important points that were previously under-explained, and they guided us in improving both the clarity and completeness of the revised manuscript.
>
> ## Weakness 1
>
> > The claimed O(log n) acceleration lacks formal assumptions and a proof or bound, so the theory feels anecdotal.
>
> We acknowledge that our original discussion of the $O(\log n)$ acceleration may not have clearly emphasized that this complexity can only be achieved under idealized conditions. In the revised version, we have made this explicit by clarifying that the $O(\log n)$ characterization reflects an ideal-case upper bound rather than a runtime guarantee in all practical scenarios.
>
> Intuitively, within a decoding block of length $L$, our divide-and-conquer structure follows a recursive pattern of
> (i) decoding tokens in each sub-decoding area,
> (ii) using the newly decoded tokens as anchors to split each area into smaller sub-areas, and
> (iii) recursing on these sub-areas.
> In the idealized case where, at each iteration, every sub-decoding area successfully decodes at least one internal token that splits the area into smaller parts, the resulting recursion tree has height $O(\log L)$. In this sense, the *recursive depth* (and thus the number of decoding iterations) scales as $O(\log L)$, in contrast to $O(L)$ sequential, token-by-token decoding.
>
> That said, this argument relies on assumptions about how often internal tokens are decoded and how effectively they partition the sub-areas. In practice, these conditions may not hold perfectly, and the empirical speedup is influenced by additional factors such as model behavior and implementation details. To avoid overstating the theoretical guarantee, we revised the wording to use a more cautious description.

---

> > ### Author Response · Authors · 2025-11-27
> > **Response to Reviewer 4: Page 2**
> >
> > ## Weakness 2
> >
> > > The KL analysis hinges on argmax→one-hot “ground truth,” which may not generalize beyond code/math settings.
> >
> > Our KL analysis does not assume that the true underlying data distribution is intrinsically one-hot. Instead, we follow the standard practice in language model training and evaluation: for each position, we take the observed token (equivalently, the argmax over the empirical data distribution) as a discrete target. This is the same surrogate used in cross-entropy training for virtually all autoregressive and diffusion-style LMs, and it is not specific to code or math domains; it applies to any discrete prediction task where a single reference token is available.
> >
> > To further support that our analysis and method are not restricted to code/math scenarios, we have added experiments on IF-Eval, which involves non-code, non-math instruction-following tasks. The results on IF-Eval show that the proposed decoding strategy and its KL-based motivation generalize well beyond code and mathematical reasoning, reinforcing that our use of argmax-based targets is a domain-agnostic modeling choice rather than a code-specific assumption.
> >
> > **Main results of Hierarchy-dLLM on the LLaDA-Instruct-8B model and LLaDA-1.5-8B model across five benchmarks.**
> > We report task performance (**Accuracy Score**) and decoding efficiency. Efficiency is measured by **TPS** (throughput per second), reflecting practical throughput, and **TPF** (tokens per forward call), indicating how many tokens are decoded per model invocation. Values in parentheses denote the relative **performance change** compared to the baseline and the **speedup factor** with respect to decoding efficiency. The best performance and highest TPS/TPF are highlighted in bold.
> >
> > | Task | Method | LLaDA-Instruct-8B<br>Score ↑ | LLaDA-Instruct-8B<br>TPF ↑ | LLaDA-Instruct-8B<br>TPS ↑ | LLaDA-1.5-8B<br>Score ↑ | LLaDA-1.5-8B<br>TPF ↑ | LLaDA-1.5-8B<br>TPS ↑ |
> > | :--- | :--- | :---: | :---: | :---: | :---: | :---: | :---: |
> > | **GSM8K** | Vanilla | 77.26 | 0.52 | 2.35 | 83.17 | 0.65 | 1.99 |
> > | | Fast-dLLM | **77.86(0.6+)** | 2.85(5.48×) | 12.80(5.45×) | 83.32(0.15+) | 3.10(4.77×) | 9.54(4.79×) |
> > | | WINO | 68.75(8.51−) | 3.24(6.23×) | 10.95(4.66×) | 81.25(1.92−) | 3.97(6.10×) | 8.00(4.02×) |
> > | | Hierarchy-dLLM | 77.18(0.08−) | **3.79(7.29×)** | **19.41(8.26×)** | **83.70(0.53+)** | **4.25(6.54×)** | **14.83(7.45×)** |
> > | **Math500** | Vanilla | 41.20 | 0.83 | 7.87 | 39.80 | 0.84 | 7.96 |
> > | | Fast-dLLM | 40.60(0.6−) | 2.71(3.27×) | 24.99(3.17×) | 39.40(0.4−) | 2.79(3.32×) | 25.81(3.24×) |
> > | | WINO | 37.40(3.8−) | 3.78(4.55×) | 30.73(3.90×) | 41.00(1.2+) | 3.75(4.46×) | 30.48(3.83×) |
> > | | Hierarchy-dLLM | **41.60(0.4+)** | **3.53(4.25×)** | **37.34(4.74×)** | **41.60(1.8+)** | **3.99(4.75×)** | **42.25(5.31×)** |
> > | **Humaneval** | Vanilla | 43.90 | 0.93 | 8.56 | 43.29 | 0.93 | 8.56 |
> > | | Fast-dLLM | 43.90(0+) | 2.94(3.16×) | 27.12(3.17×) | 42.07(1.22−) | 2.97(3.19×) | 27.45(3.21×) |
> > | | WINO | 43.90(0+) | 3.80(4.08×) | 28.15(3.29×) | 42.68(0.61−) | 3.84(4.12×) | 28.53(3.33×) |
> > | | Hierarchy-dLLM | **44.51(0.61+)** | **3.93(4.23×)** | **41.52(4.85×)** | **45.12(1.83+)** | **4.20(4.52×)** | **44.18(5.16×)** |
> > | **MBPP** | Vanilla | 37.60 | 0.13 | 0.65 | 40.20 | 0.16 | 0.80 |
> > | | Fast-dLLM | 37.60(0+) | 1.73(10.81×) | 7.26(11.17×) | **40.40(0.2+)** | 1.73(10.81×) | 8.40(10.5×) |
> > | | WINO | **38.40(0.8+)** | 1.27(9.76×) | 4.56(7.02×) | 34.80(5.4−) | 1.37(8.56×) | 4.92(6.15×) |
> > | | Hierarchy-dLLM | 37.60(0+) | **2.03(15.62×)** | **11.20(17.23×)** | **40.40(0.2+)** | **2.29(14.31×)** | **12.70(15.88×)** |
> > | **IF-Eval** | Vanilla | **57.67** | 0.59 | 6.28 | 58.78 | 0.59 | 6.29 |
> > | | Fast-dLLM | 57.30(0.37−) | 1.45(2.46×) | 15.46(2.46×) | 58.23(0.55+) | 1.38(2.34×) | 14.43(2.29×) |
> > | | WINO | 55.64(2.03−) | 1.64(2.78×) | 14.33(2.28×) | **58.60(0.18−)** | 1.59(2.69×) | 13.95(2.22×) |
> > | | Hierarchy-dLLM | 57.12(0.45−) | **1.82(3.08×)** | **22.14(3.53×)** | **59.15(0.37+)** | **1.74(2.95×)** | **21.21(3.37×)** |
> >
> > ## Weakness 3
> >
> > > Minor citation issue Line146: the text still shows “MMaDA (?)” and should be fixed to the actual reference.
> >
> > Thanks for catching this problem. The “MMaDA (?)” placeholder at Line 146 was unintended, and we have corrected it to the proper citation in the revised version.

---

### Official Review · Reviewer_YBgB · 2025-10-30

**Soundness:** 2
**Presentation:** 4
**Contribution:** 2
**Rating:** 2
**Confidence:** 4

**Summary:**

This paper proposes a new decoding method for dllm. They decode a block in a divide-and-conquer manner. Specifically, they recursively partition masked spans into sub-decoding areas. Within each area, tokens are unmasked based on confidence scores. In this way, dllm can generate multiple tokens per step without harming accuracy and maintain stability by avoiding dense consecutive undecoded spans.

**Strengths:**

The method has a clear motivation. The KL divergence study comparing sparse vs. consecutive masking is convincing and well-presented, grounding the method in a measurable phenomenon.

Empirical results show that the proposed decoding method achieves significant speedup comparing to standard decoding.

The method is training-free. There is no retraining cost.

**Weaknesses:**

**Limited novelty**: The proposed method is largely a modification of the confidence-based decoding strategy.

**Performance drop**: This method improves decoding speed by forcing that at least one token is unmasked in each area in each denoising step. However, there is an assumption behind this motivation: decoding in a divide-and-conquer way won't hurt performance. This is obviously wrong. Experimental results show that this Hierarchical decoding suffers from a performance drop even with the remasking strategy (Table 2).

**Questionable scalability**: When batch > 1, the hierarchical decoding is inefficient to implement.

**Questions:**

see weaknesses.

---

> ### Author Response · Authors · 2025-11-27
> **Response to Reviewer 3: Page 1**
>
> # Response to Reviewer 3
>
> We sincerely thank you for your thoughtful evaluation of our work and for the insightful comments you provided. Your feedback has been extremely helpful in clarifying our presentation, strengthening the experiments, and better positioning our method within the rapidly evolving literature. We truly appreciate the time and care you devoted to our paper. Here comes our responses to your questions.
>
> ## Weaknesses 1
>
> > Limited novelty: The proposed method is largely a modification of the confidence-based decoding strategy.
>
> We appreciate your concerns regarding novelty and would like to clarify that our contribution goes substantially beyond a direct modification of a simple confidence-based rule. While confidence scores are indeed one ingredient, the core of our method lies in a new *divide-and-conquer decoding structure* and a set of *interacting mechanisms* that together enable adaptive, hierarchical parallel sampling for dLLMs.
>
> First, at the structural level, we introduce *adaptive, parallel sub-decoding areas* that are recursively defined by the model’s own predictions. Instead of applying a global confidence rule on a fixed mask pattern, our method organizes decoding into a hierarchy of sub-areas that are dynamically formed and refined through the subdivision stage, and employs coordinated mechanisms trying to realize a theoretical speedup of $O(\log n)$.
> This hierarchical partitioning is not present in prior confidence-based schemes like Fast-dLLM, which typically rely on static or coarse-grained masking patterns. Our design explicitly converts the parallel decoding challenge into a sequence of more tractable sub-problems, enabling principled hierarchical parallelism rather than merely adjusting a threshold on flat confidence scores.
>
> Second, at the algorithmic level, our contribution is not merely introducing additional thresholds, but designing a cohesive "best-effort" decoding strategy tailored for the hierarchical structure.
> Unlike simple confidence truncation, our three-stage mechanism functions as an interlocking system to balance aggressive generation with strict error control. Specifically, this strategy ensures that each sub-decoding area contributes maximum progress while strictly committing only reliable tokens and dynamically rectifying early deviations to prevent error propagation in the recursive process.
> Crucially, our method is not sensitive to specific hyperparameter values. As demonstrated in the ablation studies (Figure 3a and Figure 3b), Hierarchy-dLLM maintains consistent performance improvements across a wide range of settings for these parameters. This indicates that the efficiency gains stem from the structural advantage of the decoding strategy itself, rather than fine-tuned hyperparameter overfitting.
>
> Third, experimentally, we show that this structural and algorithmic design leads to substantially stronger speed–accuracy trade-offs than existing confidence-based parallel samplers. Across extensive benchmarks, Hierarchy-dLLM significantly outperforms both Fast-dLLM, which relies solely on confidence thresholds for parallel decoding, and WINO, a complex multi-stage parallel decoding framework. Notably, our approach achieves an average speedup of nearly 50% over these baselines while maintaining comparable or superior generation quality.
>
> These empirical improvements are not explained by a simple threshold adjustment; they arise from the synergy between hierarchical decoding structure, multi-level confidence control, and remasking-based correction.

---

> > ### Author Response · Authors · 2025-11-27
> > **Response to Reviewer 3: Page 2**
> >
> > ## Weakness 2
> >
> > > Performance drop: This method improves decoding speed by forcing that at least one token is unmasked in each area in each denoising step. However, there is an assumption behind this motivation: decoding in a divide-and-conquer way won't hurt performance. This is obviously wrong. Experimental results show that this Hierarchical decoding suffers from a performance drop even with the remasking strategy (Table 2).
> >
> > We would like to respectfully clarify that we do not claim Hierarchy-dLLM achieves completely lossless acceleration across all models, nor is that the underlying assumption of our work. Instead, our primary research objective is to seek an optimal balance between decoding efficiency and generation accuracy.
> >
> > As discussed in our introduction regarding the "curse of parallel decoding," training-free parallel strategies approximate the joint distribution of tokens using marginal probabilities. It is theoretically challenging to achieve simultaneous multi-token sampling without any deviation from the greedy baseline. Therefore, a minor performance variance is an expected trade-off for achieving significant acceleration in a training-free manner.
> >
> > While some deviation is inevitable, our Preliminary Study (Section 2) empirically demonstrates that the "sparse masking" pattern used in Hierarchy-dLLM induces significantly lower distributional drift (KL divergence) compared to consecutive masking. This confirms that while our divide-and-conquer approach cannot entirely eliminate the drop, it effectively mitigates it, offering a more robust approximation than naive parallel decoding.
> >
> > The performance drop observed in Table 2 is largely attributable to the model architecture. Unlike LLaDA (a diffusion-native model), Dream-7B is adapted from an Autoregressive backbone, resulting in inherently weaker parallel capabilities. This is evidenced by the fact that the baseline Fast-dLLM also suffers a performance drop on this model. However, Hierarchy-dLLM still achieves a superior speed-accuracy trade-off compared to Fast-dLLM.
> >
> > Sacrificing a small margin of accuracy for improved efficiency is a common strategy to bridge the gap between theoretical performance and practical application constraints. In future work, we plan to explore post-training adaptations to explicitly align the model’s distribution with the hierarchical decoding behavior, which we expect will further converge the accuracy gap.
> >
> > ## Weakness 3
> >
> > > Questionable scalability: When batch > 1, the hierarchical decoding is inefficient to implement.
> >
> > Hierarchical decoding is not difficult or inefficient to batch in our setting. Our method is designed to be implemented entirely with standard batched tensor operations in PyTorch, and batching simply amounts to adding an extra batch dimension.
> >
> > In practice, we represent token states, masks, confidence scores, and sub-decoding areas as tensors with shape `[batch size, length, vocabulary size]`. All key steps of Hierarchy-dLLM (high-/low-threshold selection, area-wise argmax, subdivision, and remasking) are implemented using vectorized PyTorch matrix operations (e.g., batched `argmax`, masked updates, and batched index/mask manipulation), without per-sample Python loops.
> >
> > To substantiate this, we have implemented a batched version of Hierarchy-dLLM and included it in the anonymous code repository. Our experiments with this batched implementation show that the relative speedups reported in the paper still hold under batched inference, indicating that the hierarchical divide-and-conquer structure is fully compatible with efficient GPU batching in practice.

---

### Official Review · Reviewer_Hwju · 2025-10-31

**Soundness:** 3
**Presentation:** 3
**Contribution:** 3
**Rating:** 6
**Confidence:** 3

**Summary:**

This paper proposes Hierarchy-dLLM, a novel decoding framework designed to address the challenge of parallel decoding in discrete Diffusion Large Language Models (dLLMs). Due to the so-called “curse of parallel decoding,” dLLMs often revert to greedy decoding, resulting in inference efficiency that lags behind autoregressive (AR) models. The authors first conduct a preliminary study revealing that the spatial distribution of masked tokens has a significant impact on decoding stability: consecutive masked regions induce large distribution shifts, whereas sparsely distributed masked tokens maintain lower KL divergence and improve prediction stability. Building on this observation, Hierarchy-dLLM adopts a divide-and-conquer strategy in a training-free manner. It begins by treating a large masked region as the initial decoding area and iteratively decodes tokens based on confidence scores. Newly decoded tokens serve as anchors to split the remaining masked regions into smaller, independent sub-decoding areas, on which the same process is recursively and parallelly applied until all tokens are decoded. Experiments conducted on dLLM models such as LLaDA and Dream, across mathematical and code generation benchmarks, demonstrate that Hierarchy-dLLM achieves up to 17× speedup over vanilla decoding and about 1.5× acceleration compared to Fast-dLLM.

**Strengths:**

1. Significant efficiency improvement: The proposed training-free decoding strategy leads to a remarkably large speedup in inference efficiency.
2. Clear and insightful motivation: The experiments clearly demonstrate that sparse masking significantly reduces KL divergence compared to continuous masking, providing a strong empirical foundation for the proposed approach.
3. Comprehensive experiments: The paper presents fairly extensive experiments that demonstrate the generality and applicability of the proposed method.

**Weaknesses:**

1. Hyperparameter sensitivity: The proposed method introduces three key threshold hyperparameters that require grid search within a certain range. This raises concerns about the method’s robustness to hyperparameter choices and whether a single configuration can perform well across all datasets. Moreover, in Figure 3, the explored hyperparameter range appears relatively narrow; a broader tuning analysis would be necessary to better illustrate the method’s stability and generality.
2. Some results indicate that the acceleration comes at a substantial cost to performance, as seen in many entries of Tables 2, 5, and 6.
3. Beyond the empirical observations, the authors should provide a clearer explanation of why spatial sparsity constitutes a superior strategy, either from an empirical justification or an intuitive theoretical perspective.
4. This appears to be a rapidly evolving field, and it would be beneficial to include a more detailed discussion of concurrent works on parallel sampling in the final version, for example in the appendix.

**Questions:**

See Weaknesses

---

> ### Author Response · Authors · 2025-11-27
> **Response to Reviewer 2: Page 1**
>
> # Response to Reviewer 2
>
> Thanks for your time and valuable feedback. We have carefully considered all comments and revised the manuscript accordingly.
>
> ## Weakness 1
>
> > Hyperparameter sensitivity: The proposed method introduces three key threshold hyperparameters that require grid search within a certain range. This raises concerns about the method’s robustness to hyperparameter choices and whether a single configuration can perform well across all datasets. Moreover, in Figure 3, the explored hyperparameter range appears relatively narrow; a broader tuning analysis would be necessary to better illustrate the method’s stability and generality.
>
> We appreciate the your comments regarding hyperparameter sensitivity and the scope of our analysis. We would like to clarify that our method is designed to be robust without requiring extensive grid search for practical deployment, and we have expanded our analysis to demonstrate this stability over a broader range.
>
> We have established a robust default setting for the LLaDA-1.5-8B model: $\tau_{high} = 0.8$, $\tau_{low} = 0.5$, and $\tau_{remask} = 0.01$.
>
> In our new experiments presented in Table 1, we applied this fixed default configuration across all benchmarks. The results show that the "Single Configuration" performance is highly competitive with the per-task tuned results, with little loss in accuracy or generation speed. This confirms that Hierarchy-dLLM is robust and practically deployable without the need for the expensive grid search.
>
> Regarding the observation that the hyperparameter range in Figure 3 appeared narrow, we have added a comprehensive sensitivity analysis in the Appendix, extending the threshold sweep from 0.50 to 0.90. The results (illustrated in figure~7) reveal two key trends: The model's performance rises initially and stabilizes above 80 once the threshold exceeds 0.65. While the Tokens Per Second naturally decreases as the threshold increases, due to stricter verification, it is crucial to highlight that even the minimum TPS observed at the strictest threshold (0.90) remains significantly higher than all baselines reported in Table 1 in our paper.
>
> Finally, while the default settings are sufficient for high performance, we presented the per-task tuned results in the main paper to establish the method's theoretical upper bound. Since different datasets exhibit varying confidence distributions and difficulty levels, allowing for modest adjustments enables the model to adapt to specific domain characteristics. However, as the broader analysis confirms, this tuning is an optimization for peak efficiency rather than a requirement for functionality.
>
> **Table 1: Results of Hierarchy-dLLM on LLaDA-1.5-8B model across five benchmarks with same hyper parameters.**
>
> | Task | Method | Score ↑ | TPF ↑ | TPS ↑ |
> | :--- | :--- | :--- | :--- | :--- |
> | **GSM8K** | Vanilla | **83.17** | 0.65 | 1.99 |
> | | Hierarchy-dLLM | 82.81(0.36$-$) | **4.10(6.31$\times$)** | **14.12($7.10\times$)** |
> | **Math500** | Vanilla | **39.80** | 0.84 | 7.96 |
> | | Hierarchy-dLLM | 38.00(1.8$-$) | **4.03(4.80$\times$)** | **45.61(5.73$\times$)** |
> | **Humaneval** | Vanilla | 43.29 | 0.93 | 8.56 |
> | | Hierarchy-dLLM | **45.12(1.83$+$)** | **4.20(4.52$\times$)** | **44.18(5.16$\times$)** |
> | **MBPP** | Vanilla | 40.20 | 0.16 | 0.80 |
> | | Hierarchy-dLLM | **40.40(0.2$+$)** | **2.30(14.38$\times$)** | **12.70(15.88$\times$)** |
> | **IF-Eval** | Vanilla | 58.78 | 0.59 | 6.29 |
> | | Hierarchy-dLL | **59.15(0.37$+$)** | **1.74(2.95$\times$)** | **21.21(3.37$\times$)** |

---

> ### Author Response · Authors · 2025-11-27
> **Response to Reviewer 2: Page 2**
>
> ## Weakness 2
>
> > Some results indicate that the acceleration comes at a substantial cost to performance, as seen in many entries of Tables 2, 5, and 6.
>
> We agree that aggressive acceleration can lead to some loss in performance, especially for training-free methods, and we do not claim that divide-and-conquer decoding is lossless in all settings.
>
> Our position is:
>
> 1.  Our goal is a favorable speed–accuracy trade-off, not zero-drop. As discussed in our introduction regarding the "curse of parallel decoding," training-free parallel strategies approximate the joint distribution of tokens using marginal probabilities. It is theoretically challenging to achieve simultaneous multi-token sampling without any deviation from the greedy baseline. Targeting real-world applications where minor performance drops are acceptable for significant latency reduction, Hierarchy-dLLM aims for a favorable speed–accuracy trade-off rather than strictly zero degradation. While no training-free method achieves high acceleration without loss in complex settings, our approach delivers superior trade-offs compared to existing baselines, and in certain diffusion-native cases, even improves performance while accelerating decoding.
> 2.  Hierarchy-dLLM demonstrates robust efficacy across different architectures. For diffusion-native models ,such as LLaDA-Instruct-8B and LLaDA-1.5-8B, it leverages alignment with diffusion-style decoding to achieve substantial speedups (up to ~17×) with comparable or improved accuracy. Meanwhile, for autoregressive-origin models like Dream-7B where parallelization typically incurs some accuracy cost, Hierarchy-dLLM maintains a superior speed–accuracy trade-off, with acceleration outperforming baselines like Fast-dLLM.
>
> ## Weakness 3
>
> > Beyond the empirical observations, the authors should provide a clearer explanation of why spatial sparsity constitutes a superior strategy, either from an empirical justification or an intuitive theoretical perspective.
>
> Thanks for the suggestion. Spatial sparsity constitutes a superior strategy because it fundamentally resolves the conflict between parallel generation and the conditional independence assumption. As illustrated in our Introduction, naive parallel decoding over consecutive spans often leads to incoherent repetition—such as generating "pens, pens, and pens" instead of "pens, papers, and books"—because the model attempts to predict adjacent tokens simultaneously without knowing their neighbors. By enforcing a sparse distribution of masked tokens, we effectively break these strong local dependencies, transforming a blind block-generation problem into a set of well-constrained "cloze" tasks. This allows the model to leverage its bidirectional attention mechanism to attend to unmasked "anchor" tokens scattered throughout the sequence, thereby ensuring semantic consistency.
>
> This intuition is strongly supported by our Preliminary Study, which quantifies the stability of different masking patterns. Our experiments demonstrate that while consecutive masking leads to a rapid increase in KL divergence (distributional drift) as the number of tokens grows, sparse masking maintains a consistently low divergence that closely approximates step-by-step generation. Essentially, spatial sparsity minimizes the uncertainty in parallel decoding by maximizing the contextual information available to each masked position, allowing the model to achieve significant speedups without the performance degradation typically associated with multi-token prediction
>
> In addition to the cross-entropy surrogate for KL divergence used in the preliminary study, we have further analyzed the effect of sparse masking versus consecutive masking under the same one-step generation setup in Figure~6.
> To be explicit, we compare Sparse Mask and Consecutive Mask using the same approximate KL divergence protocol as in the preliminary study section, and now they are evaluated directly against the standard ground-truth sequences rather than step-by-step results generated by dLLM.  The new results are reported in Figure 6 of the appendix. They are consistent with our original findings: under identical conditions, sparse masking leads to lower surrogate KL, which means closer alignment to the ground-truth distribution, than consecutive masking.
>
> This additional analysis supports the sparse mask hypothesis from a complementary angle and shows that the benefit of sparse masking is not tied to a particular metric choice, but also holds when we measure divergence with respect to the actual dataset targets.

---

> > ### Author Response · Authors · 2025-11-27
> > **Response to Reviewer 2: Page 3**
> >
> > ## Weakness 4
> >
> > > This appears to be a rapidly evolving field, and it would be beneficial to include a more detailed discussion of concurrent works on parallel sampling in the final version, for example in the appendix.
> >
> > Thanks for this suggestion. In the revised version, we have substantially expanded the Related Work section to include a more detailed discussion of recent advances in acceleration of dLLMs.
> > Our new version of Related Work section is:
> >
> > ### Diffusion Large Language Models
> >
> > Diffusion modeling for language has recently gained significant attention. Early work extended score matching to discrete spaces with score entropy to build Score Entropy Discrete Diffusion models. To simplify and generalize this paradigm, masked diffusion formulations reinterpreted objectives as weighted cross-entropy losses, leading to improved training and flexibility. Scaling studies further demonstrated that Masked Diffusion Models follow comparable scaling laws to autoregressive methods and enable efficient compute–quality trade-offs.
> >
> > At larger scale, diffusion has been positioned as a credible foundation model alternative: LLaDA shows diffusion-based LLMs trained with pretraining and fine-tuning can rival strong AR Models like LLaMA and even surpass GPT‑4o on specific tasks. Extensions to robotics and multimodality include LLaDA-VLA, which adapts diffusion VLMs for vision-language-action policy learning , and MMaDA , which introduces a unified multimodal diffusion architecture with shared reasoning and generation abilities . Complementary to the above, DREAM explores discrete reasoning in autoregressive models to improve controllability and reasoning quality.
> >
> > ### Acceleration Techniques for DLLMs
> >
> > Recent work on efficient inference for dLLMs has rapidly expanded along two main axes: (i) parallel decoding strategies, and (ii) caching techniques that reduce the cost of bidirectional attention and KV storage. A number of system-level frameworks further integrate these algorithmic advances into unified inference stacks.
> >
> > #### Parallel Decoding Strategies
> >
> > To accelerate dLLM inference, researchers leverage token confidence. Fast-dLLM proposes a training-free, confidence-aware parallel decoding for dLLMs. CreditDecoding utilizes "Trace Credits" based on historical logits to accelerate the convergence of underconfident tokens. LocalLeap employs local determinism propagation around high-confidence anchors to reduce decoding steps. Dimple applies a confident decoding strategy dynamically adjusting token generation counts to significantly reduce iterations.
> > Several approaches introduce verification steps to ensure quality while accelerating inference. WINO introduces a revokable decoding mechanism that aggressively drafts tokens and uses bidirectional context to re-mask and refine errors. Adaptive Parallel Decoding inverts standard speculation by mixing dLLM marginals with a small autoregressive model to dynamically adjust sample sizes. Other works focus on sampling dynamics and model structure. SlowFast Sampling alternates between exploratory and accelerated decoding stages based on certainty and convergence principles. Addressing fixed-length limitations, dLLM-Var enables native variable generation lengths by training the model to predict [EOS] tokens.
> >
> > #### KV Cache Optimization
> >
> > Given the bidirectional attention costs in dLLMs, efficient cache management is critical.
> > Fast-dLLM introduces a block-wise approximate KV cache tailored for bidirectional models, enabling cache reuse with negligible performance drop alongside confidence-aware decoding. DLLM-Cache proposes a training-free framework that combines long-interval prompt caching with partial response updates guided by feature similarity.
> > To address memory constraints, Sparse-dLLM leverages persistent cross-layer sparsity to dynamically evict unimportant tokens while retaining pivotal ones, maintaining performance with reduced memory usage. D$^2$Cache reuses KV states for stable tokens via a dual adaptive cache. MaskKV leverages mask-token–guided scoring and adaptive budgeting for fine-grained KV compression.

---

### Official Review · Reviewer_hjHW · 2025-11-09

**Soundness:** 2
**Presentation:** 2
**Contribution:** 2
**Rating:** 6
**Confidence:** 3

**Summary:**

This paper introduces Hierarchy-dLLM, a training-free parallel decoding strategy for dLLMs. The method is motivated by a preliminary study suggesting that decoding is more stable when the remaining masked tokens are sparsely distributed rather than in a contiguous block. Based on this, the proposed algorithm employs a divide-and-conquer approach: it recursively partitions contiguous masked spans into smaller, independent "sub-decoding areas" by decoding high-confidence tokens that act as separators. This process is governed by a set of confidence-based heuristics and thresholds to decide which tokens to decode in each step. The authors evaluate their method on several dLLMs and benchmarks, reporting significant speedups (up to 17x) over vanilla decoding and outperforming the Fast-dLLM baseline in both speed and, at times, accuracy.

**Strengths:**

- Novel Motivation: The preliminary study (Section 2.2) highlighting the benefit of sparse masks is an interesting and insightful contribution that provides a new perspective on the challenges of parallel decoding.

- Creative Algorithmic Design: The application of a divide-and-conquer strategy to dLLM decoding is a novel and clever idea.

- Strong Reported Speedups: The method achieves very large reductions in the number of forward passes (high TPF) and significant improvements in wall-clock time (high TPS), demonstrating its potential for practical impact.

- Training-Free: The approach does not require any model fine-tuning, making it readily applicable to any pre-trained dLLM.

**Weaknesses:**

- Heuristic Complexity and Hyperparameter Burden: The elegance of the "divide-and-conquer" concept is undermined by the complexity of the decoding rules in Section 3.2. The method depends on a carefully orchestrated set of three thresholds, and the ablation study (Figure 3) shows that TPS, a key metric, is quite sensitive to the choice of `τ_high`. A "training-free" method that requires expensive, task-specific hyperparameter tuning is of limited practical value. The paper does not provide a principled way to set these thresholds, which is a major weakness.

- Incomplete and Potentially Weak Baselines: The paper's primary parallel decoding baseline is Fast-dLLM. However, recent literature includes other advanced training-free samplers (e.g., those based on revokable decoding like WINO, or dynamic strategies like SlowFast). By not comparing against a broader set of state-of-the-art methods, the paper's claims of superiority are not fully substantiated. It is unclear if Hierarchy-dLLM is truly better or just better than Fast-dLLM.

- Potentially Flawed Foundational Study: The entire motivation rests on the preliminary study in Section 2.2. The use of a one-hot approximation for the stepwise distribution and the resulting cross-entropy surrogate for KL divergence is a reasonable but non-trivial simplification. More importantly, this setup defines "ground truth" as the model's own sequential output, which is somewhat circular. While the trend shown is plausible, the study is not rigorous enough to serve as conclusive proof, making the foundation of the method less solid than presented.

- Algorithmic Overhead is Not Analyzed: The method requires managing a dynamic set of sub-decoding areas, finding the max confidence within each area, re-partitioning, and potentially re-masking. This introduces non-negligible computational overhead on the CPU side that is not present in simpler samplers. While TPS implicitly includes this, the paper does not discuss or analyze this overhead, making it difficult to assess how the method would scale in a highly optimized inference engine.

**Questions:**

1.  Can you provide a more principled method or at least a robust heuristic for setting the three thresholds (`τ_high`, `τ_low`, `τ_remask`)? How much performance is lost if a single "default" set of thresholds is used across all tasks and models, compared to the per-task tuned values used in the paper?

2.  To strengthen the paper's claims, could you provide a comparison against at least one other recent, sophisticated training-free baseline from the literature on a key benchmark like GSM8K?

3.  The pseudocode in Algorithm 1 appears to have a logical error. The two `if` conditions for decoding `t*` seem redundant and potentially contradictory. Could you please clarify the exact logic for when a token within a sub-area is decoded? For example, is it decoded if *either* condition is met, or is there a sequence of checks?

4.  Regarding the preliminary study, have you considered alternative metrics besides the cross-entropy surrogate for KL divergence to validate the "sparse mask" hypothesis? For instance, does the token-level prediction accuracy (against ground truth) improve more under sparse masking than consecutive masking?

---

> ### Author Response · Authors · 2025-11-27
> **Response to Reviewer 1: Page 1**
>
> Thanks for your response. We sincerely appreciate your encouraging remarks about our contributions as well as the detailed critiques you provided. Here comes our responses to your questions.
>
> ---
>
> ## Response to Question 1
>
> > Can you provide a more principled method or at least a robust heuristic for setting the three thresholds ($\tau_{high}$, $\tau_{low}$, $\tau_{remask}$)? How much performance is lost if a single "default" set of thresholds is used across all tasks and models, compared to the per-task tuned values used in the paper?
>
> Regarding the selection of the three thresholds ($\tau_{high}, \tau_{low}, \tau_{remask}$), our principled strategy is to base our choices on a set of robust default parameters, making adjustments only to explore the method's upper bounds.
>
> For the LLaDA-1.5-8B model, we established the default configuration as $\tau_{high} = 0.8$, $\tau_{low} = 0.5$, and $\tau_{remask} = 0.01$. In our newly added experiments (Table 1), we demonstrate that applying these fixed default parameters across all benchmarks yields performance that is highly competitive with the per-task tuned results. The loss in accuracy and generation speed is minimal, confirming that Hierarchy-dLLM is practically deployable without the need for expensive hyperparameter searches.
>
> This stability is further corroborated by our previous ablation studies in Figure 3(a) and (b). The results indicate that the model is highly tolerant to variations around these default values. When $\tau_{high}$ and $\tau_{low}$ are varied within a reasonable range around the default settings, the resulting accuracy fluctuations remain within a narrow margin of approximately $\pm 2\%$. This proves that the method does not rely on fragile, precise tuning to function effectively.
>
> Despite the effectiveness of the default configuration, we presented per-task tuned results in the main paper to demonstrate the upper bound of the method’s potential. Different tasks and datasets inherently exhibit distinct difficulty levels and calibration patterns, leading to varying confidence distributions. By allowing for modest, task-specific adjustments, we can better align the thresholds with these specific distributions to maximize the speed–accuracy trade-off. Thus, while the default settings provide a robust solution, the tuned results highlight the peak efficiency achievable when adapting to specific domain characteristics.
>
> ### Table 1. Results of Hierarchy-dLLM on LLaDA-1.5-8B model across five benchmarks with same hyper parameters.
>
> | **Task**   | **Method**       | **Score ↑**       | **TPF ↑**              | **TPS ↑**                |
> |-----------|------------------|-------------------|------------------------|--------------------------|
> | GSM8K     | Vanilla          | **83.17**         | 0.65                   | 1.99                     |
> |           | Hierarchy-dLLM   | 82.81 (0.36−)     | **4.10 (6.31×)**       | **14.12 (7.10×)**        |
> | Math500   | Vanilla          | **39.80**         | 0.84                   | 7.96                     |
> |           | Hierarchy-dLLM   | 38.00 (1.8−)      | **4.03 (4.80×)**       | **45.61 (5.73×)**        |
> | Humaneval | Vanilla          | 43.29             | 0.93                   | 8.56                     |
> |           | Hierarchy-dLLM   | **45.12 (1.83+)** | **4.20 (4.52×)**       | **44.18 (5.16×)**        |
> | MBPP      | Vanilla          | 40.20             | 0.16                   | 0.80                     |
> |           | Hierarchy-dLLM   | **40.40 (0.2+)**  | **2.30 (14.38×)**      | **12.70 (15.88×)**       |
> | IF-Eval   | Vanilla          | 58.78             | 0.59                   | 6.29                     |
> |           | Hierarchy-dLLM   | **59.15 (0.37+)** | **1.74 (2.95×)**       | **21.21 (3.37×)**        |

---

> > ### Author Response · Authors · 2025-11-27
> > **Response to Question 1: Page 2**
> >
> > ---
> > ## Question 2
> >
> > > To strengthen the paper's claims, could you provide a comparison against at least one other recent, sophisticated training-free baseline from the literature on a key benchmark like GSM8K?
> >
> > We have added a direct comparison with a more recent and sophisticated training-free baseline, WINO [1] decoding, under the same experimental conditions (see Table 2).
> > To be concrete, we use the official implementation provided by WINO’s author and migrate WINO into the lm-eval framework to keep the evaluation metric aligned with other experiments, and evaluate it on all five benchmarks using the hyperparameters recommended in the paper.
> >
> > Across these datasets, Hierarchy-dLLM consistently achieves a better overall speed–accuracy trade-off: for comparable or higher speedups, our method maintains equal or higher task performance, while WINO tends to suffer larger accuracy degradation at similar acceleration levels.
> >
> > These results indicate that Hierarchy-dLLM is competitive not only with Fast-dLLM but also with more advanced training-free decoding strategies, and that its hierarchical design provides stronger combined benefits in both efficiency and performance preservation.

---

> > > ### Author Response · Authors · 2025-11-27
> > > **Response to Question 1: Page 3**
> > >
> > > ### Table 2. Main results of Hierarchy-dLLM on the LLaDA-Instruct-8B model and LLaDA-1.5-8B model.
> > > We report task performance (**Accuracy Score**) and decoding efficiency. Efficiency is measured by **TPS** (throughput per second), reflecting practical throughput, and **TPF** (tokens per forward call), indicating how many tokens are decoded per model invocation. The best performance and highest TPS/TPF are highlighted in bold.
> > >
> > > | **Task**  | **Method**       | **LLaDA-Instruct-8B Score ↑** | **LLaDA-Instruct-8B TPF ↑** | **LLaDA-Instruct-8B TPS ↑** | **LLaDA-1.5-8B Score ↑** | **LLaDA-1.5-8B TPF ↑** | **LLaDA-1.5-8B TPS ↑** |
> > > |-----------|------------------|-------------------------------|-----------------------------|-----------------------------|--------------------------|------------------------|------------------------|
> > > | GSM8K     | Vanilla          | 77.26                         | 0.52                        | 2.35                        | 83.17                    | 0.65                   | 1.99                   |
> > > |           | Fast-dLLM        | **77.86 (0.6+)**              | 2.85 (5.48×)                | 12.80 (5.45×)               | 83.32 (0.15+)            | 3.10 (4.77×)           | 9.54 (4.79×)           |
> > > |           | WINO             | 68.75 (8.51−)                 | 3.24 (6.23×)                | 10.95 (4.66×)               | 81.25 (1.92−)            | 3.97 (6.10×)           | 8.00 (4.02×)           |
> > > |           | Hierarchy-dLLM   | 77.18 (0.08−)                 | **3.79 (7.29×)**            | **19.41 (8.26×)**           | **83.70 (0.53+)**        | **4.25 (6.54×)**       | **14.83 (7.45×)**      |
> > > | Math500   | Vanilla          | 41.20                         | 0.83                        | 7.87                        | 39.80                    | 0.84                   | 7.96                   |
> > > |           | Fast-dLLM        | 40.60 (0.6−)                  | 2.71 (3.27×)                | 24.99 (3.17×)               | 39.40 (0.4−)             | 2.79 (3.32×)           | 25.81 (3.24×)          |
> > > |           | WINO             | 37.40 (3.8−)                  | 3.78 (4.55×)                | 30.73 (3.90×)               | 41.00 (1.2+)             | 3.75 (4.46×)           | 30.48 (3.83×)          |
> > > |           | Hierarchy-dLLM   | **41.60 (0.4+)**              | **3.53 (4.25×)**            | **37.34 (4.74×)**           | **41.60 (1.8+)**         | **3.99 (4.75×)**       | **42.25 (5.31×)**      |
> > > | Humaneval | Vanilla          | 43.90                         | 0.93                        | 8.56                        | 43.29                    | 0.93                   | 8.56                   |
> > > |           | Fast-dLLM        | 43.90 (0+)                    | 2.94 (3.16×)                | 27.12 (3.17×)               | 42.07 (1.22−)            | 2.97 (3.19×)           | 27.45 (3.21×)          |
> > > |           | WINO             | 43.90 (0+)                    | 3.80 (4.08×)                | 28.15 (3.29×)               | 42.68 (0.61−)            | 3.84 (4.12×)           | 28.53 (3.33×)          |
> > > |           | Hierarchy-dLLM   | **44.51 (0.61+)**             | **3.93 (4.23×)**            | **41.52 (4.85×)**           | **45.12 (1.83+)**        | **4.20 (4.52×)**       | **44.18 (5.16×)**      |
> > > | MBPP      | Vanilla          | 37.60                         | 0.13                        | 0.65                        | 40.20                    | 0.16                   | 0.80                   |
> > > |           | Fast-dLLM        | 37.60 (0+)                    | 1.73 (10.81×)               | 7.26 (11.17×)               | **40.40 (0.2+)**         | 1.73 (10.81×)          | 8.40 (10.5×)           |
> > > |           | WINO             | **38.40 (0.8+)**              | 1.27 (9.76×)                | 4.56 (7.02×)                | 34.80 (5.4−)             | 1.37 (8.56×)           | 4.92 (6.15×)           |
> > > |           | Hierarchy-dLLM   | 37.60 (0+)                    | **2.03 (15.62×)**           | **11.20 (17.23×)**          | **40.40 (0.2+)**         | **2.29 (14.31×)**      | **12.70 (15.88×)**     |
> > > | IF-Eval   | Vanilla          | **57.67**                     | 0.59                        | 6.28                        | 58.78                    | 0.59                   | 6.29                   |
> > > |           | Fast-dLLM        | 57.30 (0.37−)                 | 1.45 (2.46×)                | 15.46 (2.46×)               | 58.23 (0.55+)            | 1.38 (2.34×)           | 14.43 (2.29×)          |
> > > |           | WINO             | 55.64 (2.03−)                 | 1.64 (2.78×)                | 14.33 (2.28×)               | **58.60 (0.18−)**        | 1.59 (2.69×)           | 13.95 (2.22×)          |
> > > |           | Hierarchy-dLLM   | 57.12 (0.45−)                 | **1.82 (3.08×)**            | **22.14 (3.53×)**           | **59.15 (0.37+)**        | **1.74 (2.95×)**       | **21.21 (3.37×)**      |

---

> ### Author Response · Authors · 2025-11-27
> **Response to Question 1: Page 4**
>
> ---
>
> ## Quetsion 3
>
> > The pseudocode in Algorithm 1 appears to have a logical error. The two if conditions for decoding t\* seem redundant and potentially contradictory. Could you please clarify the exact logic for when a token within a sub-area is decoded? For example, is it decoded if either condition is met, or is there a sequence of checks?
>
> Thank you for pointing this out. The apparent contradiction comes from a typo in the pseudocode: the roles of $\tau_{\text{high}}$ and $\tau_{\text{low}}$ were accidentally swapped in one of the conditions. We have corrected this in the revised version. After fixing the typo, the two conditions are no longer redundant or contradictory, and the pseudocode now matches the description in the main text.
>
> Our decoding strategy operates within each sub-decoding area using a progressively relaxed, confidence-based rule. At each step, we first commit all tokens whose maximum posterior probability exceeds a high threshold, enabling parallel, high-precision updates. If no token in an area reaches this threshold, we still decode the single most confident token as long as it surpasses a lower threshold, ensuring local progress without relying on extremely uncertain predictions. If an entire iteration would otherwise produce no updates, we fall back to decoding the globally most confident token to guarantee steady advancement. Finally, before repartitioning, we remask any previously decoded token whose confidence has dropped below a remask threshold, allowing the model to revise dubious earlier decisions. This procedure balances efficiency and reliability by decoding aggressively when confidence is high, conservatively when it is moderate, and correcting low-confidence tokens as context evolves.
>
> ---
>
> ## Quetsion 4
>
> > Regarding the preliminary study, have you considered alternative metrics besides the cross-entropy surrogate for KL divergence to validate the "sparse mask" hypothesis? For instance, does the token-level prediction accuracy (against ground truth) improve more under sparse masking than consecutive masking?
>
> Thanks for the suggestion. In addition to the cross-entropy surrogate for KL divergence used in the preliminary study, we have further analyzed the effect of sparse masking versus consecutive masking under the same one-step generation setup in Figure 6 based on your advice.
> To be explicit, we compare Sparse Mask and Consecutive Mask using the same approximate KL divergence protocol as in the preliminary study section, but now they are evaluated directly against the standard ground-truth sequences rather than step-by-step results generated by dLLM.  Notably, the plots demonstrate a clear diverging trend: as the number of masked tokens increases, the performance gap between the two strategies widens substantially to more than 60 on both benchmark, highlighting the superior stability of sparse masking as the generation burden increases. They are consistent with our original findings that under identical conditions, sparse masking leads to lower surrogate KL, which means closer alignment to the ground-truth distribution, than consecutive masking.
>
> ---
>
> ## Concern on Algorithmic Overhead
>
> > Algorithmic Overhead is Not Analyzed: The method requires managing a dynamic set of sub-decoding areas, finding the max confidence within each area, re-partitioning, and potentially re-masking. This introduces non-negligible computational overhead on the CPU side that is not present in simpler samplers. While TPS implicitly includes this, the paper does not discuss or analyze this overhead, making it difficult to assess how the method would scale in a highly optimized inference engine.
>
> Thanks for raising this insightful point regarding the algorithmic overhead on the CPU side.
> We implement Hierarchy-dLLM using efficient PyTorch tensor operations, so most computations run entirely on GPU without incurring heavy CPU-side control overhead. To quantify the runtime cost of our hierarchical selection logic, we benchmark both vanilla diffusion decoding and Hierarchy-dLLM on HumanEval using a single H20 GPU. As shown in Figure 8 in our appendix, for LLaDA-Instruct-8B the token-selection and index-manipulation procedures in Hierarchy-dLLM account for only 1.2% of the end-to-end inference time, an increase of merely 0.5% compared to vanilla decoding, while the model forward pass remains the dominant cost. At the same time, Hierarchy-dLLM reduces the average number of function evaluations (NFE) from 128 to 64 and nearly halves the total decoding latency. This confirms that our method introduces negligible overhead beyond model forwards and can efficiently accelerate dLLM inference in practice.
>
> [1] Wide-In, Narrow-Out: Revokable Decoding for Efficient and Effective DLLMs. 2025

---

### Author Response · Authors · 2025-11-27

Thanks to all reviewers for your thoughtful and constructive feedback. We are encouraged that the reviewers consider efficient inference for diffusion LLMs to be important and timely.

To address the concerns raised, we have conducted significant additional experiments and revisions:

1. Robustness: We tested a fixed "default" set of hyperparameters across benchmarks to prove stability (Response to Reviewer1, Reviewer2).
2. New Baselines: We added comparisons with WINO[1] and other state-of-the-art methods (Response to Reviewer1).
3. New Benchmark: We added results on IF-Eval to demonstrate effectiveness on non-code/math tasks (Response to Reviewer4).
4. New Pre-study: We added a study comparing "Sparse Mask" vs. "Ground Truth" to validate our foundational hypothesis (Response to Reviewer1, Reviewer2).
5. We added a runtime breakdown separating model forward time and decoding-algorithm overhead introduced by hierarchy-dLLM. (Response to Reviewer1)
6. Batching: We implemented and evaluated Hierarchy-dLLM with Batch Size > 1 (Response to Reviewer3).

[1] Wide-In, Narrow-Out: Revokable Decoding for Efficient and Effective DLLMs. 2025.

---

### Meta-Review · Area_Chair_aMRc · 2026-01-10

**Summary:**

Hierarchy-dLLM introduces a tree-structured hierachical decoding strategy and yields promising empirical results. Three reviewers have suggested acceptance. I have read the paper and also find it a sound and useful method. The authors also provided comprehensive rebuttal, which in my opinion, have addressed most of the the concerns. Therefore, I recommand acceptance.

**Reviewer Concerns:**

Most of the concerns are about empirical validation, and I think the authors did a good job in the rebuttal stage.

**Reviewer Scores:**

Three reviewers recommand acceptance and one reviewer remained rejection (with no further interaction with the authors).

---

### Decision · Program_Chairs · 2026-01-26

Accept (Poster)